# An $\alpha$-regret analysis of Adversarial Bilateral Trade

**Yossi Azar**
Department of Computer Science
Tel Aviv University, Israel
azar@tauex.tau.ac.il

**Amos Fiat**
Department of Computer Science
Tel Aviv University, Israel
fiat@tau.ac.il

**Federico Fusco**
Department of Computer, Control
and Management Engineering
Sapienza Università di Roma, Italy
fuscof@diag.uniroma1.it

## Abstract

We study sequential bilateral trade where sellers and buyers valuations are completely arbitrary (*i.e.*, determined by an adversary). Sellers and buyers are strategic agents with private valuations for the good and the goal is to design a mechanism that maximizes efficiency (or gain from trade) while being incentive compatible, individually rational and budget balanced. In this paper we consider gain from trade which is harder to approximate than social welfare.

We consider a variety of feedback scenarios and distinguish the cases where the mechanism posts one price and when it can post different prices for buyer and seller. We show several surprising results about the separation between the different scenarios. In particular, we show that (a) it is impossible to achieve sublinear $\alpha$-regret for any $\alpha < 2$, (b) but with full feedback sublinear 2-regret is achievable (c) with a single price and partial feedback one cannot get sublinear $\alpha$ regret for any constant $\alpha$ (d) nevertheless, posting two prices even with one-bit feedback achieves sublinear 2-regret, and (e) there is a provable separation in the 2-regret bounds between full and partial feedback.

## 1 Introduction

The bilateral trade problem arises when two rational agents, a seller and a buyer, wish to trade a good; they both hold a private valuation for it, and their goal is to maximize their utility. The solution to the problem consists in designing a mechanism that intermediates between the two parties to make the trade happen. Ideally, the mechanism should maximize social welfare even though the agents act strategically (*incentive compatibility*) and should guarantee non-negative utility to the agents (*individual rationality*). Furthermore, we are interested in mechanisms for bilateral trade that do not subsidize the agents (*budget balance*). Obvious mechanisms that satisfy incentive compatibility, individual rationality, and budget balanced, are posted price mechanisms. Two common metrics are used to measure the efficiency of a mechanism: social welfare subsequent to trade and gain from trade (*i.e.*, the increase in social welfare). Consider a mechanism that posts prices $p$ (price for the seller) and $q$ (price for the buyer) to agents with valuations $s$ and $b$, formally we have:

- **Social Welfare**: $\mathrm{SW}(p, q, s, b) = s + (b - s) \cdot \mathbb{I}\{s \leq p \leq q \leq b\}$ [1]
- **Gain from trade**: $\mathrm{GFT}(p, q, s, b) = (b - s) \cdot \mathbb{I}\{s \leq p \leq q \leq b\}$

---

[1] We use $\mathbb{I}\{Q\}$ for the indicator variable that takes the value 1 if the predicated $Q$ is true and zero otherwise.

36th Conference on Neural Information Processing Systems (NeurIPS 2022).

It is clear from these expressions that if we are interested in exact optimality then maximizing gain from trade is equivalent to maximizing social welfare. However, Myerson and Satterthwaite [1983] showed that there is *no mechanism* for bilateral trade that is simultaneously social welfare maximizing (alternately, gain from trade maximizing), incentive compatible, individually rational, and budget balanced[2]. It follows that the best one can hope for is an incentive compatible, individually rational, and budget balanced mechanism that approximates the optimal social welfare. This creates an asymmetry between the two metrics: a multiplicative $c$ approximation to the maximal gain from trade implies an approximation at least as good ($\geq c$) to the maximal social welfare but not vice versa. Ergo, it is harder to approximate the gain from trade than to approximate social welfare. For example, consider an instance where the seller has valuation $0.99$ and the buyer is willing to pay up to $1$: irrespective of if a trade occurs or not, $99\%$ of the optimal social welfare is guaranteed. *I.e.*, a mechanism that posts a price of zero and generates no trade gets a good approximation to the social welfare. Contrariwise, the gain from trade is non-zero only if the mechanism manages to post prices in the narrow $[0.99, 1]$ interval.

The vast body of work subsequent to Myerson and Satterthwaite [1983] primarily considers the Bayesian version of the problem, where agents' valuations are drawn from some distribution and the efficiency is evaluated in expectation with respect to the valuations' randomness. There are many incentive compatible mechanisms that give a constant approximation to the social welfare (in the Bayesian setting), e.g., see Blumrosen and Dobzinski [2014]. On the other hand, finding a constant approximation to the gain from trade has been a long standing problem and only a very recent paper of Deng et al. [2022] has given a Bayesian incentive compatible mechanism for this problem. In this paper, we deal with the harder scenario where an adversary determines seller and buyer valuations (*i.e.*, valuations are not drawn from some distribution). Ergo, positive results in the Bayesian setting are inapplicable in our setting.

Following Cesa-Bianchi et al. [2021a], we consider the sequential adversarial bilateral trade problem, where at each time step $t$, a new seller-buyer pair arrives. The seller has some private valuation $s_t \in [0, 1]$ representing the smallest price she is willing to accept; conversely, the buyer holds as private information $b_t \in [0, 1]$, *i.e.*, the largest price she is willing to pay to get the good. Concurrently, the mechanism posts price $p_t$ to the seller and $q_t$ to the buyer. If they both accept ($s_t \leq p_t$ and $q_t \leq b_t$), then the trade happens at those prices, otherwise the agents leave forever. By the requirement that the mechanism is budget balanced, the prices posted by the mechanism are such that $p_t \leq q_t$. At the end of each time step, the mechanism receives some feedback that depends on the outcome of the trade. Ideally, we would like to have a strategy for the sequential bilateral trade problem whose average gain from trade converges to that of the best fixed posted price mechanism in hindsight. However, as Cesa-Bianchi et al. [2021a] showed, this is a hopeless task.

Our goal in this work is then to achieve mechanisms whose average performance converges to a constant factor of the best fixed posted price mechanism in hindsight. We would like to find the smallest $\alpha \geq 1$ such that the $\alpha$-regret [Kakade et al., 2009] is sublinear in the time horizon $T$:

$$\max_{p,q} \sum_{t=1}^{T} \text{GFT}(p, q, s_t, b_t) - \alpha \cdot \mathbb{E}\left[\sum_{t=1}^{T} \text{GFT}(p_t, q_t, s_t, b_t)\right].$$

If the goal is only to maximize gain from trade, there is never any sense in offering two different prices (to the seller and buyer). However, critically, offering two prices is provably helpful in the context of a learning algorithm.

To conclude the description of our learning framework, we specify the type of feedback received by the mechanism. We focus on the two extremes of the feedback spectrum. On the one hand, we study the full feedback model, where, after prices are posted, the mechanism learns both seller and buyer valuations $(s_t, b_t)$. On the other hand, we investigate a more realistic *partial feedback* model, the *one-bit feedback*, where the learner only discovers if a trade took place or not. We also consider an intermediate (partial feedback) model, called the *two-bit feedback model*. In this model, the learner posts (one or two) prices, and learns if the buyer is willing to trade and if the seller is willing to trade, at these prices. Clearly, a trade actually occurs only if both are willing to trade. Note that these two models enforce the desirable property that buyers and sellers only communicate to the mechanism a minimal amount of information useful for the trade, without disclosing their actual valuations.

---

[2]This impossibility result holds even when the (private) agents valuations are assumed to be drawn from some (public) random distributions and the incentive compatibility is only enforced in expectation.

| | Full Feedback | Two-bit feedback | One-bit feedback |
|---|---|---|---|
| Single price | $O(\sqrt{T})$ - Theorem 2 | $\Omega(T)$ - Theorem 4 | |
| Two prices | $\Omega(\sqrt{T})$ - Theorem 3 | $\Omega(T^{2/3})$ - Theorem 6 | $O(T^{3/4})$ - Theorem 5 |

Table 1: Summary of 2-regret results in various settings.

## 1.1 Overview of our Results

We present our results for the adversarial sequential bilateral trade problem (see also Table 1).

- We show that no learning algorithm can achieve sublinear $\alpha$-regret for any $\alpha < 2$ (Theorem 1). This holds in the full feedback model (and thus for both partial feedback models).
- We give a learning algorithm with full feedback that achieves $\tilde{O}(\sqrt{T})$ 2-regret[3] (Theorem 2) and show that no algorithm can improve upon this (Theorem 3).
- We show that if limited to a single price, no learning algorithm achieves sublinear $\alpha$-regret for any constant $\alpha$ in either partial feedback model: one or two-bit feedback (Theorem 4).
- Given the negative results above, we show that allowing the learning algorithm to post two prices gives sublinear 2-regret even for one-bit feedback (Theorem 5). This means that our learning algorithm achieves, on average, at least half of the gain from trade of the best fixed price in hindsight, using only one bit of feedback at each step!
- We show a separation between partial versus full feedback by giving a $\Omega(T^{2/3})$ lower bound in the former model on the 2-regret for any learning algorithm (Theorem 6).

The gaps in Table 1 may appear misleading because upper bounds in weaker models apply in stronger models and lower bounds in stronger models apply in weaker models. The only remaining open gap in our results is between the $\Omega(T^{2/3})$ lower bound and the $O(T^{3/4})$ upper bound that hold for two prices and partial feedback (either one or two-bit feedback). It is also worth noting that in our worst case model, two prices are required but one-bit suffices for sublinear 2-regret. This is a different qualitative behavior than the one observed in the stochastic case [Cesa-Bianchi et al., 2021a], where it is enough to use one single price but the two-bit feedback is required to achieve sublinear (1-)regret. One may wonder why two prices are helpful at all in our adversarial setting, given their suboptimality in maximizing the gain from trade. It turns out that by randomizing over two prices it is possible to estimate the (non-stochastic) valuations of the agents.

## 1.2 Technical challenges

**From experts to prices.** As already observed in Cesa-Bianchi et al. [2021a], the full feedback model nicely fits into the prediction with experts framework [Cesa-Bianchi and Lugosi, 2006]: there is a clear mapping between expert and prices and the mechanism can easily reconstruct the gain that each price/expert experiences using the feedback received. The main challenge here is given by the continuous nature of the possible prices, as the usual experts framework assumes a finite number of experts. There are workarounds that exploit some regularity of the gain function such as the Lipschitz property or convexity/concavity [see, e.g., Cesa-Bianchi and Lugosi, 2006, Hazan, 2016, Slivkins, 2019]. Unfortunately, gain from trade is not such a function. Moreover, in our adversarial setting, we cannot adopt the smoothing trick used in Cesa-Bianchi et al. [2021a], where they assume some regularity on the agents' distribution to argue that $\mathbb{E}[\mathrm{GFT}(\cdot)]$ becomes Lipschitz. Our main technical tool to address this issue is a discretization claim that allows us to compare the performance of the best fixed price in $[0, 1]$ with that of the best on a finite grid.

**A magic estimator.** Consider any of the two partial feedback models; there at each time step $t$ the learner only receives minimal information about what happened at time $t$: namely, one or two-bit versus the full knowledge of $\mathrm{GFT}_t(\cdot)$. Note that this type of feedback is strictly more difficult than the classic bandit feedback [Cesa-Bianchi and Lugosi, 2006], where the learner always observes at least the gain its action incurred. Our main technical tool to circumvent this issue is given by the design of a procedure that, posting two randomized prices, is able to estimate the GFT in a given price. This unbiased estimator is then used in a carefully designed block decomposition of the time horizon to achieve sublinear 2-regret in presence of this very poor feedback.

---

[3]The $\tilde{O}$ hides poly-logarithmic terms

**Lower bounds.** For our lower bounds we adopt two different strategies. In Theorems 1 and 4 we construct randomized instances where no algorithm can learn anything: the only prices the learner could use to discriminate between different instances are cautiously hidden, while all the other prices do not reveal any useful information, given the type of feedback considered. The randomized instances used in Theorems 3 and 6 involve instead a more structured approach; given the challenge posed by the contemporary handling of the multiplicative and additive part of the 2-regret. To this end, we carefully hide the optimal ex-post prices and make it hard for the learner to achieve small (1-)regret with respect to some "second best" prices. A crucial task we often face is to "hide" some small finite sets of critical prices from the learning algorithm. We employ two techniques to do so: random shifts (as in the proof of Theorem 4) and repeatedly dividing overlaps (Theorems 1 and 3).

## 1.3 Related work

The work most closely related to ours is Cesa-Bianchi et al. [2021a]. There, the authors study the same sequential bilateral trade problem as we do, with the objective of minimizing the (1-)regret with respect to the best fixed price. They focus on the (easier) stochastic model, where the adversary chooses a distribution over valuations and not a deterministic sequence like we do. A full characterization of the minimax regret regimes is offered, for the same type of feedback we consider (the one-bit feedback is only addressed in the extended version [Cesa-Bianchi et al., 2021b][4]) and with various regularity assumptions on the underlying random distributions. Cesa-Bianchi et al. [2021a] also provide the first result for the adversarial setting we consider, showing that no learning algorithm can achieve sublinear 1-regret. Regret minimization in the context of economics has been studied in many papers [e.g., Morgenstern and Roughgarden, 2015, Cesa-Bianchi et al., 2015, Ho et al., 2016, Daskalakis and Syrgkanis, 2016, Lykouris et al., 2016]. In particular, Kleinberg and Leighton [2003] studied the one-sided pricing problem, proving a $\tilde{O}(T^{2/3})$ upper bound on the regret in the adversarial setting and opening a fruitful line of research [Blum et al., 2004, Blum and Hartline, 2005, Bubeck et al., 2019]. The notion of $\alpha$-regret has been formally introduced by Kakade et al. [2009] but was already present in Kalai and Vempala [2005]. It has then found applications in linear [Garber, 2021] and submodular optimization [Roughgarden and Wang, 2018], learning with sleeping actions [Emamjomeh-Zadeh et al., 2021], combinatorial auctions [Roughgarden and Wang, 2019] and market design [Niazadeh et al., 2021]. We mention that our work fits in the line of research that studies online learning with feedback models different from full information and the bandit ones; our one and two-bit feedback models share similarities with the feedback graphs model (see e.g., Alon et al. [2017], van der Hoeven et al. [2021], Esposito et al. [2022]) and the partial monitoring framework (see e.g., Bartók et al. [2014], Lattimore and Szepesvári [2019]).

While Myerson and Satterthwaite [1983] were the first to thoroughly investigate the bilateral trade problem in the Bayesian setting with their famous impossibility result, it was the seminal paper of Vickrey [Vickrey, 1961] that introduced the problem, proving that any mechanism that is welfare maximizing, individually rational, and incentive compatible may not be budget balanced. In the Bayesian setting, it was only very recently that Deng et al. [2022] gave the first (Bayesian) incentive compatible, individually rational, and budget balanced mechanism achieving a constant factor approximation of the optimal gain from trade. Prior to this paper, a posted price $O\left(\log \frac{1}{r}\right)$ approximation bound was achieved by [Colini-Baldeschi et al., 2017], with $r$ being the probability that a trade happens (i.e., the value of the buyer is higher than the value of the seller). The literature also includes many individually rational, incentive compatible, and budget balanced mechanisms achieving a constant factor approximation of the optimal social welfare. Blumrosen and Dobzinski [2014] proposed a simple posted price mechanism, the median mechanism, yielding a 2-approximation of the optimal social welfare; the same authors then implemented a randomized fixed price mechanism improving the approximation to $e/(e-1)$ [Blumrosen and Dobzinski, 2021]. Recently, Dütting et al. [2021] showed that even posting one single sample from the seller distribution as price is enough to achieve a 2 approximation to the optimal social welfare. The class of fixed price mechanism is of particular interest as it has been shown that all (dominant strategy) incentive compatible and individually rational mechanisms that enforce a stricter notion of budget balance, i.e., the so-called

---

[4]Cesa-Bianchi et al. [2021b] use a definition of gain from trade that is slightly different than ours when the prices posted to seller and buyer differ. In practice, however, the two definitions are equivalent. In particular, all our lower bounds apply automatically to their definition, while the analyses of our learning algorithms carry over exactly the same also with their definition.

---
**Learning Protocol of Sequential Bilateral Trade**

---
  **for** time $t = 1, 2, \ldots$ **do**
      a new seller/buyer pair arrives with (hidden) valuations $(s_t, b_t) \in [0, 1]^2$
      the learner posts prices $p_t, q_t \in [0, 1]$
      the learner receives a (hidden) reward $\mathrm{GFT}_t(p_t, q_t) := \mathrm{GFT}(p_t, q_t, s_t, b_t) \in [0, 1]$
      a feedback $z_t$ is revealed

---

strong budget balance (where the mechanism is not allowed to subsidize or extract revenue from the agents) are indeed fixed price [Hagerty and Rogerson, 1987, Colini-Baldeschi et al., 2016].

## 2 Preliminaries

The formal protocol for the sequential bilateral trade follows Cesa-Bianchi et al. [2021a]. At each time step $t$, a new pair of seller and buyer arrives, each with private valuations $s_t$ and $b_t$ in $[0, 1]$; the learner posts two prices: $p_t \in [0, 1]$ to the seller and $q_t \in [0, 1]$ to the buyer. A trade happens if and only if both agents agree to trade, i.e., when $s_t \le p_t$ and $q_t \le b_t$. Since we want our mechanism to enforce budget balance, we require that $p_t \le q_t$ for all $t$. When a trade occurs, the learner is awarded the resulting increase in social welfare, i.e., $b_t - s_t$. The learner then observes some feedback $z_t$. The gain from trade at time $t$ depends on the valuations $s_t$ and $b_t$ and on the price posted. To simplify the notation we introduce the following:

$$\mathrm{GFT}_t(p, q) := \mathrm{GFT}(p, q, s_t, b_t) = \mathbb{I}\{s_t \le p \le q \le b_t\} \cdot (b_t - s_t)$$

When the two prices are equal, we omit one of the arguments to simplify the notation.

Given any constant $\alpha \ge 1$, the $\alpha$-regret of a learning algorithm $\mathcal{A}$ against a sequence of valuations $\mathcal{S}$ on time horizon $T$ is defined as follows

$$R_T^\alpha(\mathcal{A}, \mathcal{S}) := \max_{p, q \in [0,1]^2} \sum_{t=1}^T \mathrm{GFT}_t(p, q) - \alpha \cdot \sum_{t=1}^T \mathbb{E}\left[\mathrm{GFT}_t(p_t, q_t)\right].$$

In the right side of the equation, the dependence on $\mathcal{S}$ is contained in the $\mathrm{GFT}_t(\cdot)$. Note that the expectation in the previous formula is with respect to the internal randomization of the learning algorithm: $p_t$ and $q_t$ are the (possibly random) prices posted by $\mathcal{A}$.

The $\alpha$-regret of a learning algorithm $\mathcal{A}$, without specifying the dependence of the sequence, is defined as its $\alpha$-regret against the "worst" sequence of valuations: $R_T^\alpha(\mathcal{A}) := \sup_{\mathcal{S}} R_T^\alpha(\mathcal{A}, \mathcal{S})$. Stated differently, the performance of an algorithm is measured against an oblivious adversary that generates the sequence of valuations ahead of time: the learner has to perform well on *all* possible sequences. In this paper we study the minimax $\alpha$-regret, $R_T^{\alpha,\star}$, that measures the performance of the best (learning) algorithm versus the optimal fixed price in hindsight, on the worst possible instance: $R_T^{\alpha,\star} := \inf_{\mathcal{A}} R_T^\alpha(\mathcal{A})$. The set of learning algorithms we consider depends on which of the various settings we are dealing with. In this paper we consider a variety of such settings (*i.e.*, how many prices are posted, what feedback is available, see below Sections 2.1 and 2.2).

### 2.1 Single Price *vs.* Two Prices — Seller price and Buyer price

We consider two families of learning algorithms, differing in the nature of the probe they perform, corresponding to two notions of what it means to be budget balanced:

**Single price mechanisms.** If we want to enforce a stricter notion of budget balance, namely strong budget balance, the mechanism is neither allowed to subsidize nor extract revenue from the system. This is modeled by imposing $p_t = q_t$, for all $t$. If $p_t = q_t$ we use the notation $\mathrm{GFT}_t(p_t)$ to represent the gain from trade at time $t$.

**Two price mechanisms.** If we require that the mechanism enforces (weak) budget balance, it can post two different prices, $p_t$ to the seller and $q_t$ to the buyer, as long as $p_t \le q_t$. *I.e.*, we only require that the mechanism never subsidize a trade, we do not require that the mechanism not make a profit. In this setting we use the notation $\mathrm{GFT}_t(p_t, q_t)$ to represent the gain from trade at time $t$.

**Observation 1.** *Note that the only reason to post two prices is to obtain information. For any pair of prices $(p, q)$ with $p < q$ posting any single price $\pi \in [p, q]$ guarantees no less gain from trade.*

In particular, any budget balanced algorithm that knows the future and seeks to maximize gain from trade while repeatedly posting the same prices will never choose two different prices.

## 2.2 Feedback models

We consider three types of feedback, presented here in increasing order of difficulty for the learner. (Note that full feedback "implies" two-bit feedback which in turn implies one-bit feedback):

**Full feedback.** In the full feedback model, the learner receives both seller and buyer valuations, immediately after posting prices: formally, $z_t = (s_t, b_t)$. E.g., both seller and buyer send sealed bids that are opened immediately after the [one or two] price[s] are revealed. By Observation 1, in the full feedback model there is no reason to post two prices, as all the relevant information is revealed anyway.

**Two-bit feedback.** In two-bit feedback the algorithm observes separately if the two agents agree on the given price, *i.e.*, the feedback at time $t$ is $z_t = (\mathbb{I}\{s_t \leq p_t\}, \mathbb{I}\{q_t \leq b_t\})$.

**One-bit feedback** The one-bit feedback is arguably the minimal feedback the learner could get: the only information revealed is whether the trade occurred or not, *i.e.*, $z_t = \mathbb{I}\{s_t \leq p_t \leq q_t \leq b_t\}$.

## 2.3 Regret due to discretization

Our first theoretical result concerns the study of how discretization impacts the regret. In particular, we compare the performance of the best fixed price taken from the continuous set $[0, 1]$ to that of the best fixed price chosen from some discrete grid $Q \subset [0, 1]$. Optimizing over a continuous set may seemingly be a problem because our object, gain from trade, is discontinuous (thus non-Lipschitz), non-convex and non-concave; one cannot use the "standard approach" that makes use of such regularity conditions. What we show in the following Claim (whose proof is deferred to the Appendix together with all the other missing proofs) is that it is possible to compare the performance of the best continuous fixed price with *twice* that of the best fixed price on the grid.

**Claim 1** (Discretization error). *Let* $Q = \{q_0 = 0 \leq q_1 \leq q_2 \cdots \leq q_n = 1\}$ *be any finite grid of prices in* $[0, 1]$ *and let* $\delta(Q)$ *be the largest difference between two contiguous prices, i.e.,* $\max_{i=1,\ldots,n} |q_i - q_{i-1}|$, *then for any sequence* $\mathcal{S} = (s_1, b_1), \ldots, (s_T, b_T)$ *and any price* $p$ *we have*

$$\sum_{t=1}^{T} \mathrm{GFT}_t(p) \leq 2 \cdot \max_{q \in Q} \sum_{t=1}^{T} \mathrm{GFT}_t(q) + \delta(Q) \cdot T.$$

*Let* $\mathcal{A}$ *be any learning algorithm that posts prices* $(p_t, q_t)$, *then the following inequality holds:*

$$R_T^2(\mathcal{A}) \leq 2 \sup_{\mathcal{S}} \left\{ \max_{q \in Q} \sum_{t=1}^{T} \mathrm{GFT}_t(q) - \sum_{t=1}^{T} \mathbb{E}\left[\mathrm{GFT}_t(p_t, q_t)\right] \right\} + \delta(Q) \cdot T. \tag{1}$$

Before moving to the next section, we spend some words to compare our discretization result with the one in Cesa-Bianchi et al. [2021a] (Second decomposition Lemma). There the authors exploit the stochastic nature of the valuations to argue that $\mathbb{E}\left[\mathrm{GFT}_t(\cdot)\right]$ is Lipschitz, under some regularity assumptions on the random distributions. We study the adversarial model, thus we cannot use this "smoothing" procedure; this is why we lose an extra multiplicative factor of 2.

## 3 Full Feedback

In this section, we study the full feedback model, where the learner receives as feedback both seller and buyer valuations after posting a single price (see Observation 1). The learner can thus evaluate $\mathrm{GFT}_t(p)$ for all $p \in [0, 1]$, independently of the price posted. Even with this very rich feedback, we show that the impossibility result from Cesa-Bianchi et al. [2021a], *i.e.*, no learning algorithm achieves sublinear regret (1-regret) in the sequential bilateral trade problem, can be extended to hold for $\alpha$-regret for all $\alpha \in [1, 2)$.

**Theorem 1** (Lower bound on $(2 - \varepsilon)$-regret). *In the full-feedback model, for all* $\varepsilon \in (0, 1]$ *and horizons* $T$, *the minimax* $(2 - \varepsilon)$-regret satisfies $R_T^{2-\varepsilon,\star} \geq \frac{1}{8}\varepsilon T$.

To prove Theorem 1 we use Yao's Minimax Theorem: a randomized family of valuations sequences is constructed, with the property that any deterministic learner would suffer, on average, linear $2 - \varepsilon$ regret against it. The detailed proof is deferred to the Appendix, but we sketch here the main ideas. Specifically, any valuations sequence from the randomized family consists of (sell, buy) prices that have the form $(0, b_i)$ or $(s_i, 1)$, for some carefully designed $\{s_i\}_i$ and $\{b_i\}_i$. These sequences are generated iteratively in a way such that all realized [sell, buy] segments overlap and the next segment is chosen at random among two disjoint options $(0, b_i)$ or $(s_i, 1)$. Since all realized [sell, buy] segments overlap, there is at least one price in the intersection of all intervals: this is the optimal fixed price in hindsight. Conversely, at each time step no learner can post a price that guarantees a trade with probability greater than $1/2$, thus yielding the lower bound.

Moving to positive results, there is a clear connection between our problem in the full feedback and the prediction with experts framework [Cesa-Bianchi and Lugosi, 2006]. In particular, if we simplify the task of the learner to be competitive against *the best price in a finite grid*, we can use classical results on prediction with experts as a black box. Combining this with our discretization result (Claim 1), we show an $\tilde{O}(\sqrt{T})$ upper bound on the 2-regret. The details are postponed to the Appendix.

**Theorem 2** (Upper bound on 2-regret given full feedback). *In the full-feedback setting, there exists a learning algorithm $\mathcal{A}$ whose 2-regret, for $T$ large enough, respects $R_T^2(\mathcal{A}) \leq 5 \cdot \sqrt{T \cdot \log T}$.*

We conclude the full feedback analysis with a lower bound that shows that the previous result is tight up to a logarithmic factor: the minimax 2-regret of the full feedback problem is $\tilde{\Theta}(\sqrt{T})$.

**Theorem 3** (Lower bound on 2-regret given full feedback). *In the full-feedback model, for all horizons $T$ large enough, the minimax 2-regret satisfies $R_T^{2,\star} \geq \frac{1}{13}\sqrt{T}$.*

The proof uses once again Yao's Theorem and consists in constructing a randomized family of sequences such that any deterministic learning algorithm suffers, in expectation, a $\Omega(\sqrt{T})$ 2-regret. The detailed construction is deferred to the Appendix and it involves the careful combination of two scaled copies of the hard sequences used in the proof of Theorem 1.

# 4 Partial Feedback

In this section, we study the partial feedback models where the learner receives very limited information on the realizations of the gain from trade. Specifically, one or two bits that describe the relative positions of the prices proposed to the agents and their valuations.

## 4.1 Lower bound on $\alpha$-regret posting single price given two-bit feedback

Consider a learner that is constrained to post one single price at every iteration; the same to both seller and buyer. For this class of algorithms we show a very strong impossibility result, namely that for any constant $\alpha$, there exists no algorithm achieving sublinear $\alpha$-regret. We prove this in the two-bit feedback model and thus it trivially holds also if given one-bit feedback. The core of the lower bound construction (details are postponed to the Appendix) resides in the possibility for the adversary to hide a *large* interval between many shorter ones; a learner posting only one price will not be able to locate it using partial feedback (which consists in just *counting* the number of intervals on the left and on the right).

**Theorem 4** (Lower bound on $\alpha$-regret posting single price, two-bit feedback). *In the two-bit feedback model where the learner is allowed to post one single price, for all horizons $T \in \mathbb{N}$ and any constant $\alpha > 1$, the minimax $\alpha$-regret satisfies $R_T^{\alpha,\star} \geq \frac{1}{128\alpha^3}T$.*

## 4.2 Upper bound on the 2-regret, posting two prices and given one-bit feedback

The main result in this section is presented in Theorem 5: it is possible to achieve sublinear 2-regret with one-bit feedback (and by posting two prices). We find this to be the most surprising result in this paper. The crucial ingredient of our approach is an unbiased estimator, $\widehat{\text{GFT}}$, of the gain from trade that uses two prices and *one single bit* of feedback. This seems quite remarkable. The gain from trade is a discontinuous function composed by two different objects: the difference $(b - s)$ and

---
**Estimation procedure of GFT using two prices and one-bit feedback**

---
    **Input:** price $p$

Toss a biased coin with head probability $p$

**if** head **then** Draw $U$ u.a.r. in $[0, p]$ and set $\hat{p} \leftarrow U$, $\hat{q} \leftarrow p$

**else** Draw $V$ u.a.r. in $[p, 1]$ and set $\hat{p} \leftarrow p$, $\hat{q} \leftarrow V$

Post price $\hat{p}$ to the seller and $\hat{q}$ to the buyer and observe the one-bit feedback $\mathbb{I}\{s \leq \hat{p} \leq \hat{q} \leq b\}$

    **Return:** $\widehat{\mathrm{GFT}}(p) \leftarrow \mathbb{I}\{s \leq \hat{p} \leq \hat{q} \leq b\}$                  ▷ Unbiased estimator of $\mathrm{GFT}(p)$

---

the indicator variable $\mathbb{I}\{p \in [s, b]\}$. Both these two objects are easy to estimate *independently*, but for the gain from trade we need an estimator of their product. To estimate $\mathrm{GFT}(p)$ for any fixed price $p$, we construct an estimation procedure that considers both features at the same time: it tosses a biased coin with head probability $p$; if head, it posts price $p$ to the buyer and a price drawn u.a.r. in $[0, p]$ to the seller; if tails, it posts price $p$ to the seller and a price drawn u.a.r. in $[p, 1]$ to the buyer. The formal procedure is described in the pseudocode, while the following lemma proves that this procedure yields an unbiased estimator of the gain from trade.

**Lemma 1.** *Fix any agents' valuations $s, b \in [0, 1]$. For any price $p \in [0, 1]$, it holds that $\widehat{\mathrm{GFT}}(p)$ is an unbiased estimator of $\mathrm{GFT}(p)$: $\mathbb{E}\left[\widehat{\mathrm{GFT}}(p)\right] = \mathrm{GFT}(p)$, where the expectation is with respect to the randomness of the estimation procedure.*

*Proof.* Note that $p$ is fixed and known to the learner, $s$ and $b$ are fixed but unknown and the learner has to estimate the fixed but unknown quantity $\mathrm{GFT}(p) = \mathbb{I}\{s \leq p \leq q \leq b\} \cdot (b_t - s_t)$ using only the two-bit feedback. To analyze the expected value of $\widehat{\mathrm{GFT}}(p)$ we define two random variables:

$$X_s(p) = I_{\{s \leq U \leq p \leq b\}}, \ X_b(p) = I_{\{s \leq p \leq V \leq b\}}, \ \text{where } U \sim Unif(0, p) \text{ and } V \sim Unif(p, 1).$$

If $p \notin [s, b]$, the two random variables attain value 0 with probability 1 (and are thus both unbiased estimators of $\mathrm{GFT}(p)$ in that case). Consider now $p \in [s, b]$ and compute their expectation:

$$\mathbb{E}\left[X_s(p)\right] = \mathbb{P}\left(s \leq U \leq p \leq b\right) = \mathbb{P}\left(s \leq U\right) = \frac{p - s}{p},$$

$$\mathbb{E}\left[X_b(p)\right] = \mathbb{P}\left(s \leq p \leq V \leq b\right) = \mathbb{P}\left(V \leq b\right) = \frac{b - p}{1 - p}.$$

The estimator $\widehat{\mathrm{GFT}}(p)$ works as follows: with probability $p$ it posts prices $(U, p)$, otherwise $(p, V)$, then receives the one-bit feedback from the agents and returns it. Conditioning on the result of the toss of the biased coin it is then easy to compute the expected value of $\widehat{\mathrm{GFT}}(p)$:

$$\mathbb{E}\left[\widehat{\mathrm{GFT}}(p)\right] = p\,\mathbb{E}\left[X_s(p)\right] + (1 - p)\,\mathbb{E}\left[X_b(p)\right] = \mathbb{I}\{s \leq p \leq q \leq b\}\,(b - s) = \mathrm{GFT}(p). \quad \square$$

This estimation procedure becomes a powerful tool to estimate the gain from trade that the learner would have extracted at time $t$ posting price $p$ using randomization and *one single bit* of feedback. Note here that the possibility of posting two different prices is crucial: as we have argued in the previous section, one single price is not able to do that, even for two-bit feedback. Given the estimator $\widehat{\mathrm{GFT}}$ (actually, it consists of a family of estimators: one for each price $p$) we present our learning algorithm BLOCK-DECOMPOSITION. Similarly to what is done in Chapter 4 of Nisan et al. [2007], the learner divides the time horizon in $S$ time blocks $B_\tau$ of equal length[5] and uses as subroutine some expert algorithm $\mathcal{E}$ on a meta-instance that considers each time block as a time step and each price in a suitable grid $Q$ as an action. In each block the learner posts the same price $p_\tau$ in all but $|Q|$ time steps, where it uses $\widehat{\mathrm{GFT}}$ to estimate the total gain from trade obtained in $B_\tau$ by all prices in $Q$. The details of BLOCK-DECOMPOSITION are presented in the pseudocode, and its guarantees are detailed in the following Theorem, whose proof is deferred to the Appendix.

**Theorem 5** (Upper bound on 2-regret posting two prices, one-bit feedback)**.** *In the one-bit feedback model where the learner is allowed to post two prices, the 2-regret of* BLOCK-DECOMPOSITION *(BD) is such that $R_T^2(\mathrm{BD}) \leq 5T^{3/4}\sqrt{\log(T)}$, for appropriate choices of the expert algorithm $\mathcal{E}$, grid $Q$ and number of blocks $S$.*

---
[5]For ease of exposition we assume that $S$ divides $T$. This is without loss of generality in our case, as one can always add some dummy time steps for an additive regret of at most $T/S$.

---

BLOCK-DECOMPOSITION (BD)

1: **Input:** time horizon $T$, number of blocks $S$, grid $Q$ and expert algorithm $\mathcal{E}$
2: $\Delta \leftarrow T/S$, $K \leftarrow |Q|$
3: $B_\tau \leftarrow \{(\tau-1) \cdot \Delta + 1, \ldots, \tau \cdot \Delta\}$, for all $\tau = 1, 2, \ldots, S$
4: Initialize $\mathcal{E}$ with time horizon $S$ and $K$ actions, one for each $p_i \in Q$
5: **for** each round $\tau = 1, 2, \ldots, S$ **do**
6:      Receive from $\mathcal{E}$ the price $p_\tau$
7:      Select uniformly at random an injection $h_\tau : Q \to B_\tau$          ▷ We need $\Delta >> |Q|$
8:      **for** each round $t \in B_\tau$ **do**
9:          **if** $h_\tau(p_i) = t$ for some price $p_i$ **then**
10:              Use the estimator $\widehat{\text{GFT}}(p_i)$ at time $t$ and call its output $\widehat{\text{GFT}}_\tau(p_i)$
11:          **else:** Post price $p_\tau$ and ignore feedback
12:      Feed to $\mathcal{E}$ the estimated gains $\{\widehat{\text{GFT}}_\tau(p_i)\}_{i=1,\ldots,K}$

---

### 4.3 Lower bound on $2$-regret, posting two prices and two-bit feedback

In this section, we complement the positive results for the single price and two-bit feedback setting with a lower bound on the $2$-regret achievable in the (easier) two-price and two-bit feedback setting. This lower bound strongly depends on a powerful characterization result from the partial monitoring literature [Bartók et al., 2014] and consists in constructing a class of instances with the following structure that mimics an "hard" partial monitoring game. The $[0, 1]$ interval is divided into 4 disjoint regions, the first one is composed of a single optimal price $p^\star$, then two intervals that are candidates to be the second best after $p^\star$. The only way for the learner to actually discriminate between the two candidates and assess which is the actual second best is to post prices in the last, suboptimal region of the $[0, 1]$ interval. The construction is such that there is a multiplicative factor 2 between the gain from trade of $p^\star$ and that of the second best. For the learner it is impossible to locate the single point $p^\star$ (given the structure of the feedback), and its regret with respect to the second best prices is at least $\Omega(T^{2/3})$. The reader familiar with the learning literature would recognize the similarity of this structure to the classical revealing action problem [Cesa-Bianchi et al., 2006].

The randomized family of instances that are hard to learn for any deterministic learner is easy to describe: at the beginning, the adversary randomly and uniformly select one of the two following distributions over valuations $(s, b)$ and then draws $T$ i.i.d. samples from it:

$$\begin{cases} (0, \frac{1}{2}) & \text{with probability } \frac{1}{4} + \varepsilon \\ (\frac{1}{3}, \frac{1}{2}) & \text{with probability } \frac{1}{4} - \varepsilon \\ (\frac{1}{2}, \frac{2}{3}) & \text{with probability } \frac{1}{4} \\ (\frac{1}{2}, 1) & \text{with probability } \frac{1}{4} \end{cases} \qquad \begin{cases} (0, \frac{1}{2}) & \text{with probability } \frac{1}{4} \\ (\frac{1}{3}, \frac{1}{2}) & \text{with probability } \frac{1}{4} \\ (\frac{1}{2}, \frac{2}{3}) & \text{with probability } \frac{1}{4} - \varepsilon \\ (\frac{1}{2}, 1) & \text{with probability } \frac{1}{4} + \varepsilon \end{cases}$$

We can compute the expected performance $\mathbb{E}[GFT(p)]$ of any price $p$ against them (the first, respectively second, column corresponds to the first, respectively second, distribution)

$$\begin{cases} \frac{1}{8} + \frac{\varepsilon}{2} & \text{if } p \in [0, \frac{1}{3}) \\ \frac{1}{6} + \frac{\varepsilon}{3} & \text{if } p \in [\frac{1}{3}, \frac{1}{2}) \\ \frac{1}{3} + \frac{\varepsilon}{3} & \text{if } p = \frac{1}{2} \\ \frac{1}{6} & \text{if } p \in (\frac{1}{2}, \frac{2}{3}] \\ \frac{1}{8} & \text{if } p \in (\frac{2}{3}, 1] \end{cases} \qquad \begin{cases} \frac{1}{8} & \text{if } p \in [0, \frac{1}{3}) \\ \frac{1}{6} & \text{if } p \in [\frac{1}{3}, \frac{1}{2}) \\ \frac{1}{3} + \frac{\varepsilon}{3} & \text{if } p = \frac{1}{2} \\ \frac{1}{6} + \frac{\varepsilon}{3} & \text{if } p \in (\frac{1}{2}, \frac{2}{3}] \\ \frac{1}{8} + \frac{\varepsilon}{2} & \text{if } p \in (\frac{2}{3}, 1] \end{cases}$$

It is clear that the best price is $\frac{1}{2}$, which yields an expected gain from trade that is approximately a multiplicative factor 2 larger than the one induced by the second best price, i.e. $p \in [\frac{1}{3}, \frac{1}{2})$ or $p \in (\frac{1}{2}, \frac{2}{3}]$ depending on the instance in question. The two candidates to be the second best price are an additive $\Theta(\varepsilon)$ factor away while posting prices in $[0, \frac{1}{3}) \cup (\frac{2}{3}, 1]$ gives a constant loss. The crucial property is that the only way the learner can discriminate between the two instances is to post prices in the low gain region $[0, \frac{1}{3}) \cup (\frac{2}{3}, 1]$. For example, posting a price of $\frac{1}{3}$ the learner observes a trade with probability exactly $\frac{1}{2}$ in both the distributions, while posting 0 yields a trade with probability $\frac{1}{4} + \varepsilon$ in the first case while exactly $\frac{1}{4}$ in the second (thus allowing some learning

to happen). Moreover, the learner cannot take advantage of the possibility of posting more than one price: the only useful thing to learn is where the extra $\varepsilon$ probability is, and there is no way of doing it without suffering a constant instantaneous regret; not even with two-bits of feedback. We formalize these considerations in the following lemma, whose proof is deferred to the Appendix.

**Lemma 2.** *Consider the class of learning algorithms that can post two prices (both different from $1/2$) and receive two-bit feedback. For any $\mathcal{A}$ in this class, there exists a sequence from the family we described such that the following bound on the regret holds, for some constant $c > 0$:*

$$\max_{p \neq \frac{1}{2}} \sum_{t=1}^{T} \mathrm{GFT}(p) - \mathbb{E}\left[\sum_{t=1}^{T} \mathrm{GFT}_t(p_t, q_t)\right] \geq cT^{2/3}.$$

To conclude our lower bound, we need to show how to *hide* to the learner the price $1/2$ that is clearly optimal. It is sufficient to add a small, random, perturbation of the instance.

**Theorem 6** (Lower bound on regret for two prices and two-bit feedback). *In the two-bit feedback model where the learner is allowed to post two prices, for all horizons $T \in \mathbb{N}$, the minimax $2$-regret satisfies $R_T^{2,\star} \geq \tilde{c}T^{2/3}$ for some constant $\tilde{c}$.*

*Proof.* Let $\delta > 0$ be an arbitrarily small constant. We perturb each instance of the family we have constructed earlier in the following way: the adversary draws uniformly at random a shift $x \in [0, \delta]$, then adds it to all valuations and finally it divides them all by $1 + \delta$. The valuations are still in $[0, 1]$ and the optimal price $p^\star$ has now become $\frac{1}{2(1+\delta)} + \frac{x}{1+\delta}$. The learner has now no way of pinpointing the exact location of $p^\star$ since it is impossible to locate a specific point in $[0, \delta]$ using two-bit feedback. Finally, the addition of new, independent, random noise does not make the learning of the second best price easier, i.e., the bound of Lemma 2 holds. Thus, for any learning algorithm $\mathcal{A}$, we have:

$$\begin{aligned}
\mathbb{E}\left[R_T^2(\mathcal{A})\right] \geq & \mathbb{E}\left[\max_{p \in [0,1]} \sum_{t=1}^{T} \mathrm{GFT}_t(p) - 2\max_{p \neq p^\star} \sum_{t=1}^{T} \mathrm{GFT}_t(p)\right] \\
& + 2 \cdot \mathbb{E}\left[\max_{p \neq p^\star} \sum_{t=1}^{T} \mathrm{GFT}_t(p) - \sum_{t=1}^{T} \mathrm{GFT}_t(p_t, q_t)\right] \\
\geq & \delta\Theta(T) + cT^{2/3} \geq \tilde{c} \cdot T^{2/3}.
\end{aligned}$$

$\square$

## 5 Discussion, Extensions, and Open Problems

In this paper, we investigate the sequential bilateral trade problem with adversarial valuations. We study various feedback scenarios and consider the possibility of the mechanism to post one price vs. when it can post different prices for buyer and seller. We identify the exact threshold of $\alpha$ that allows sublinear $\alpha$-regret. We show that with partial feedback it is impossible to achieve sublinear $\alpha$-regret for any constant $\alpha$ with a single price while 2-regret is achievable with 2 prices. Finally, we show a separation in the minimax 2-regret between full and partial feedback. Although in this paper we only consider the gain from trade, our positive results trivially hold with respect to social welfare. Furthermore, by modifying our lower bound from Theorem 1 it is possible to show that sublinear $\alpha$-regret is not achievable for $\alpha < 2$ with respect to social welfare. An obvious open problem, with respect to both gain from trade and to social welfare, consists in determining the exact regret term as a function of $T$. Clearly, there is a gap in our Table of results, and the exact term is yet unclear also for social welfare. We focus on the sequential problem where at each step one buyer and one seller appear. It would be interesting to study the model where multiple buyers and multiple sellers arrive at each time step and sellers have values for their goods, buyers have values for the different goods.

## Acknowledgments and Disclosure of Funding

Yossi Azar was supported in part by the Israel Science Foundation (grant No. 2304/20). Federico Fusco was supported by the ERC Advanced Grant 788893 AMDROMA Algorithmic and Mechanism Design Research in Online Markets and MIUR PRIN grant Algorithms, Games, and Digital Markets (ALGADIMAR).

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
