# Appendix

## A    Lower bounds via Yao's Minimax Theorem

An important technical tool we use to prove our lower bounds is the well known Yao's Minimax Theorem [Yao, 1977]. In particular, we apply the easy direction of the theorem, which reads (using our terminology) as follows: the $\alpha$-regret of a randomized learner against the worst-case valuations sequence is at least the minimax regret of the optimal deterministic learner against a stochastic sequence of valuations. Formally,

$$R_T^{\alpha,\star} \geq \sup_{\mathcal{A}} \mathbb{E}\left[\max_{p,q\in[0,1]^2}\sum_{t=1}^{T}\mathrm{GFT}_t(p,q) - \alpha \cdot \sum_{t=1}^{T}\mathrm{GFT}_t(p_t,q_t)\right],$$

where the expectation is with respect to the stochastic valuation sequence $\mathcal{S}$, while $\mathcal{A}$ denotes a deterministic learner. We remark that — for the minimax theorem to be applicable — the random instance $\mathcal{S}$ has to be oblivious of the learner.

## B    Missing proof from Section 2

**Claim 1** (Discretization error). *Let $Q = \{q_0 = 0 \leq q_1 \leq q_2 \cdots \leq q_n = 1\}$ be any finite grid of prices in $[0,1]$ and let $\delta(Q)$ be the largest difference between two contiguous prices, i.e., $\max_{i=1,\dots,n}|q_i - q_{i-1}|$, then for any sequence $\mathcal{S} = (s_1,b_1),\dots,(s_T,b_T)$ and any price $p$ we have*

$$\sum_{t=1}^{T}\mathrm{GFT}_t(p) \leq 2 \cdot \max_{q\in Q}\sum_{t=1}^{T}\mathrm{GFT}_t(q) + \delta(Q) \cdot T.$$

*Let $\mathcal{A}$ be any learning algorithm that posts prices $(p_t,q_t)$, then the following inequality holds:*

$$R_T^2(\mathcal{A}) \leq 2\sup_{\mathcal{S}}\left\{\max_{q\in Q}\sum_{t=1}^{T}\mathrm{GFT}_t(q) - \sum_{t=1}^{T}\mathbb{E}\left[\mathrm{GFT}_t(p_t,q_t)\right]\right\} + \delta(Q) \cdot T. \qquad (1)$$

*Proof.* Fix any sequence of valuations $\mathcal{S}$ and let $p^\star$ be the corresponding best fixed price: $p^\star \in \arg\max_{p\in[0,1]}\sum_{t=1}^{T}\mathrm{GFT}_t(p)$. If $p^\star \in Q$, then there is nothing to prove; alternatively let $q^+$ and $q^-$ be the consecutive prices on the grid such that $p^\star \in [q^-,q^+]$. For any time $t$ where $\mathrm{GFT}_t(p^\star) > 0$, either $p^\star \in [s_t,b_t] \subseteq [q^-,q^+]$, in which case

$$\mathrm{GFT}_t(p^\star) \leq (b_t - s_t) \leq (q^+ - q^-) \leq \delta(Q),$$

or $[s_t,b_t] \cap \{q^+,q^-\} \neq \emptyset$, and therefore $\mathrm{GFT}_t(p^\star) = \max\{\mathrm{GFT}_t(q^+),\mathrm{GFT}_t(q^-)\}$. All in all, we have that, for each time $t$, the following inequality holds:

$$GFT_t(p^\star) \leq GFT_t(q^+) + GFT_t(q^-) + \delta(Q)$$

Summing up over all times $t$ we get:

$$\sum_{t=1}^{T}\mathrm{GFT}_t(p^\star) \leq \sum_{t=1}^{T}GFT_t(q^+) + \sum_{t=1}^{T}GFT_t(q^-) + \delta(Q)T \leq 2\cdot\max_{q\in Q}\sum_{t=1}^{T}\mathrm{GFT}_t(q) + \delta(Q)T. \quad (2)$$

Focus now on the second part of the claim and fix any learning algorithm $\mathcal{A}$, we have:

$$R_T^2(\mathcal{A}) = \sup_{\mathcal{S}}\left\{\sum_{t=1}^{T}GFT_t(p^\star) - 2\cdot\sum_{t=1}^{T}\mathbb{E}\left[GFT_t(p_t,q_t)\right]\right\}$$

$$\leq \sup_{\mathcal{S}}\left\{\sum_{t=1}^{T}GFT_t(p^\star) - 2\max_{q\in Q}\sum_{t=1}^{T}GFT_t(q)\right\}$$

$$+ \sup_{\mathcal{S}}\left\{2\max_{q\in Q}\sum_{t=1}^{T}GFT_t(q) - 2\sum_{t=1}^{T}\mathbb{E}\left[GFT_t(p_t,q_t)\right]\right\}$$

$$\leq T\cdot\delta(Q) + 2\sup_{\mathcal{S}}\left\{\max_{q\in Q}\sum_{t=1}^{T}GFT_t(q) - \sum_{t=1}^{T}\mathbb{E}\left[GFT_t(p_t,q_t)\right]\right\},$$

where the last inequality follows from Equation (2) that holds for all sequences.    $\square$

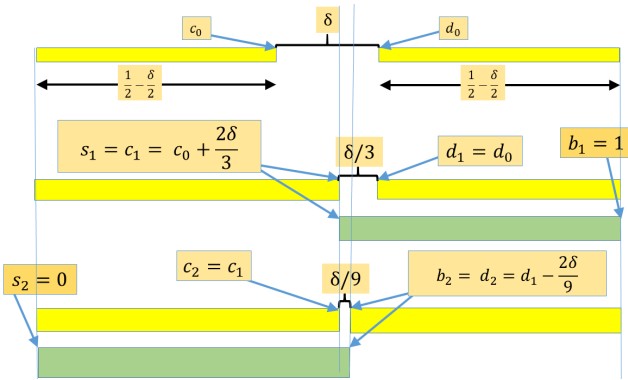

Figure 1: Lower bound construction that "hides" the optimal price.

# C  Missing proofs from Section 3

**Theorem 1** (Lower bound on $(2 - \varepsilon)$-regret). *In the full-feedback model, for all $\varepsilon \in (0, 1]$ and horizons $T$, the minimax $(2 - \varepsilon)$-regret satisfies $R_T^{2-\varepsilon,\star} \geq \frac{1}{8}\varepsilon T$.*

*Proof.* We prove this lower bound via Yao's Theorem. Fix any $\varepsilon \in (0, 1]$, we argue that there exist some constant $c_\varepsilon$ and a distribution over sequences such that the $(2 - \varepsilon)$-regret of any deterministic learning algorithm $\mathcal{A}$ against it is, on average, at least $c_\varepsilon \cdot T$. Our construction is reminiscent —and to some extent simplifies— the one given in Theorem 4.6 of Cesa-Bianchi et al. [2021a], but presents one main difference: here we construct a family of instances that is oblivious to the learner, whereas in Cesa-Bianchi et al. [2021a] they construct a single instance tailored to the learning algorithm $\mathcal{A}$. It is critical for the application of Yao's Theorem that the sequence distribution is independent of the actual algorithm. Let $\delta < \varepsilon/8$, the adversary initiates two auxiliary sequences of points $c_0 = \frac{1}{2} - \frac{1}{2}\delta$ and $d_0 = \frac{1}{2} + \frac{1}{2}\delta$ then, inductively constructs the auxiliary sequences and draws $s_{t+1}$ and $b_{t+1}$ as follows:

$$\begin{cases} c_{t+1} := c_t, \ d_{t+1} := d_t - \frac{2\delta}{3^t}, \ s_{t+1} := 0, \ b_{t+1} := d_{t+1}, & \text{with probability } 1/2 \\ c_{t+1} := c_t + \frac{2\delta}{3^t}, \ d_{t+1} := d_t, \ s_{t+1} := c_{t+1}, \ b_{t+1} := 1, & \text{with probability } 1/2. \end{cases}$$

A quick description of the procedure: at the beginning of each time step $t + 1$ the adversary has two points, $c_t$ and $d_t$, with $d_t - c_t = \delta/3^t$. Then, it chooses uniformly at random between *left* or *right*. If *left* is chosen (first line of the construction), then $c_{t+1} = c_t$, while $d_{t-1}$ is moved to the first third of the $[c_t, d_t]$ interval: $d_{t+1} = d_t - \frac{2\delta}{3^t} = c_t + \frac{\delta}{3^t}$, and the adversary posts prices $s_{t+1} = 0$ and $b_{t+1} = d_{t+1}$. If *right* is chosen, then the symmetric event happens: $d_{t+1} = d_t$, $c_{t+1}$ moves to the second third of $[c_t, d_t]$ and the adversary posts $s_{t+1} = c_{t+1}$ and $b_{t+1} = 1$. A pictorial representation of a sample run of this procedure is given in Figure 1.

At each time step the two possible realizations (for a fixed past) of the $[s_t, b_t]$ intervals are disjoint: it implies that *any price* the learner posts results in a trade with probability (over the randomness of the adversary) of at most $1/2$. As we are in a full feedback scenario, there is no point for the learner to post two prices, so we assume that $\mathcal{A}$ posts a single price.

For any realization of the randomness used in the construction of the sequence, $[s_t, b_t]$ intervals have a non-empty intersection; let $p^\star$ be some price in this intersection. Moreover, at all time steps $t$ it holds that $(b_t - s_t) \geq \frac{1}{2} - \frac{\delta}{2}$. All in all, this gives a simple bound on the total gain from trade of the best price in hindsight that holds for any realization of the valuations sequence:

$$\max_{p \in [0,1]} \sum_{t=1}^{T} \text{GFT}_t(p) = \sum_{t=1}^{T} \text{GFT}_t(p^\star) \geq \frac{T}{2} (1 - \delta).$$

Consider now what happens to the learner. We already argued that at each time step the learner obtains a trade with probability at most $1/2$. Furthermore, $(b_t - s_t) \leq \frac{1}{2} + \frac{\delta}{2}$ for all realizations.

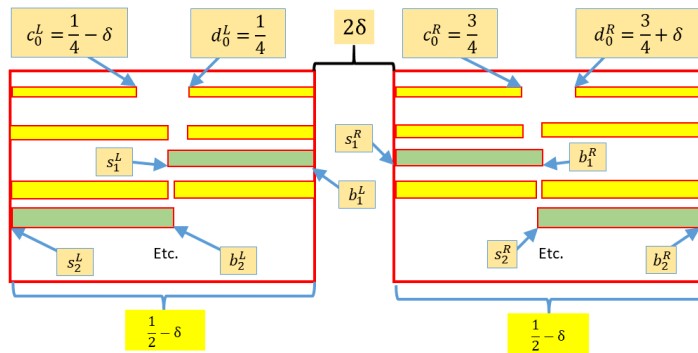

Figure 2: The proof of Theorem 3 makes use of two (appropriately scaled and shifted) copies of the lower bound from Theorem 1 (See Figure 1). In this example, the left-hand copy chooses right and then left, while the right-hand copy happened to choose left and then right. The (seller,buyer) bids at time $t$ are then chosen independently at random from $(s_t^L, b_t^L)$ and $(s_t^R, b_t^R)$.

Thus:

$$\mathbb{E}\left[\sum_{t=1}^{T} \text{GFT}_t(p_t)\right] \leq \sum_{t=1}^{T} \frac{1}{2}\left(\frac{1}{2} + \frac{\delta}{2}\right) = \frac{T}{2}\left(\frac{1}{2} + \frac{\delta}{2}\right)$$

At this point we have the desired explicit bound on the $2 - \varepsilon$ regret via Yao's Theorem:

$$R_T^{2-\varepsilon}(\mathcal{A}) \geq \frac{T}{2}(1 - \delta) - (2 - \varepsilon)\frac{T}{2}\left(\frac{1}{2} + \frac{\delta}{2}\right) = \frac{T}{4}(\varepsilon + \varepsilon\delta - 4\delta) \geq \frac{1}{8}\varepsilon T. \qquad \square$$

**Theorem 2** (Upper bound on 2-regret given full feedback). *In the full-feedback setting, there exists a learning algorithm $\mathcal{A}$ whose 2-regret, for $T$ large enough, respects $R_T^2(\mathcal{A}) \leq 5 \cdot \sqrt{T \cdot \log T}$.*

*Proof.* Consider a grid of prices $Q$ composed by $T + 1$ equally spaced points: $q_i = i/T$ for $i = 0, 1, \ldots, T$ and choose your favorite prediction with experts learning algorithm, e.g., Multiplicative Weights [Arora et al., 2012]. Given the full feedback regime, and the fact that the grid is finite, we can run an experts algorithm using as experts the points on the grid. Typically, the best experts learning algorithm exhibit a bound on the regret $O(\sqrt{T \log K})$, which becomes $O(\sqrt{T \log T})$ in our case since we have $T + 1$ experts. If we use the Multiplicative Weights algorithm against the best fixed price on the grid $Q$ with $\eta = \sqrt{\frac{\log T}{T}}$, we get by Theorem 2.5 of Arora et al. [2012]:

$$\sup_{\mathcal{S}}\left\{\max_{q \in Q}\sum_{t=1}^{T} \text{GFT}_t(q) - \sum_{t=1}^{T} \mathbb{E}\left[\text{GFT}_t(p_t)\right]\right\} \leq 2\sqrt{T \log(T + 1)}$$

Plugging this bound in Claim 1, we get the desired order of regret.

$$R_T^2(\mathcal{A}) \leq 2\sup_{\mathcal{S}}\left\{\max_{q \in Q}\sum_{t=1}^{T} \text{GFT}_t(q) - \sum_{t=1}^{T} \mathbb{E}\left[\text{GFT}_t(p_t)\right]\right\} + \delta(Q) \cdot T$$

$$\leq 4\sqrt{T \log(T + 1)} + 1 \leq 5\sqrt{T \log(T)}.$$

The first inequality is just a restatement of Equation (2) from Claim 1. The second inequality follows by combining the bound on the regret of multiplicative weight and the fact that the grid is equally spaced, thus $\delta(Q) = 1/T$. $\qquad \square$

In order to prove Theorem 3 we need the following property of Random walks.

**Lemma 3** (Property of Random Walks). *Let $S_T$ be a symmetric random walk on the line after $T$ steps, starting from $0$. Then, for $T$ large enough, it holds that $\mathbb{E}\left[|S_T|\right] \geq \frac{2}{3}\sqrt{T}$.*

*Proof.* It is well known that the expected distance of a random walk from the origin grows like $\Theta(\sqrt{T})$. Formally, the following asymptotic result holds [e.g., Palacios, 2008]

$$\lim_{T \to \infty} \frac{\mathbb{E}[|S_T|]}{\sqrt{T}} = \sqrt{\tfrac{2}{\pi}}.$$

Observe that $\sqrt{\tfrac{2}{\pi}} > 2/3$, thus there exists a finite $T_0$ such that $\mathbb{E}[|S_T|] \geq \tfrac{2}{3}\sqrt{T}$ for all $T \geq T_0$. $\quad\square$

**Theorem 3** (Lower bound on 2-regret given full feedback). *In the full-feedback model, for all horizons $T$ large enough, the minimax 2-regret satisfies $R_T^{2,\star} \geq \frac{1}{13}\sqrt{T}$.*

*Proof.* We show that there exists a distribution over valuations sequences such that any deterministic learning algorithm $\mathcal{A}$ achieves, on average, at least a 2-regret of $\frac{1}{13} \cdot \sqrt{T}$. This is enough to conclude the proof via Yao's Theorem. It may be helpful to consider Figure 2 to visualize this construction. Fix some small $\delta$ to be set later and consider two scaled copies of the lower bound construction from Theorem 1, one in $[0, \frac{1}{2} - \delta]$ and the other is $[\frac{1}{2} + \delta, 1]$. Starting from $c_0^L = \frac{1}{4} - \delta$, $d_0^L = \frac{1}{4}$, $c_0^R = \frac{3}{4}$, $d_0^R = \frac{3}{4} + \delta$, the left $L$ and right $R$ pair of sequences evolve over time and generate two distinct sequences of valuations: $(s_t^L, b_t^L) \subseteq [0, \frac{1}{2} - \delta]$ and $(s_t^R, b_t^R) \subseteq [\frac{1}{2} + \delta, 1]$. The actual sequence of valuations presented to the learner is based on these two sequences as follows: at each time step, the adversary tosses a fair coin, if it is a head, then it selects $(s_t, b_t) = (s_t^L, b_t^L)$, otherwise $(s_t, b_t) = (s_t^R, b_t^R)$. Observe that there are two independent sources of randomness in the adversary construction: the one responsible for the generation of the auxiliary sequences and the one toss of the left-right coin. We give now an upper bound on the expected performance of the learner at each time step $t$. Reasoning similarly to what we did in Theorem 1, there are four disjoint intervals of the $[0, 1]$ interval where a price could cause a trade, and each one of them is the one chosen by the adversary with probability $1/4$ ($1/2$ given by the left-right coin and another $1/2$, independently, by the evolution of the sequences $(s_t^L, b_t^L)$ and $(s_t^R, b_t^R)$). All in all, this implies that for any price the algorithm posts, it results in a trade with probability at most $1/4$. Moreover, we have the property that $(b_t - s_t) \leq (1/4 + \delta)$ at all times and for all realizations, therefore: $\mathbb{E}[\text{GFT}_t(p_t)] \leq \frac{1}{4}\left(\frac{1}{4} + \delta\right)$. We move now our attention to lower bounding the gain from trade of the best price in hindsight. Consider any realization of the sequence of coin tosses, we know that there exist two prices $p_L^\star$ and $p_R^\star$ such that $p_L^\star$ guarantees a trade in every time step where the result of the left-right coin gives left, and $p_R^\star$ does the same when the coin gives right. In addition, we know that $(b_t - s_t) \geq \frac{1}{4} - \delta$. All in all we have that, for all realizations of the randomness,

$$\sum_{t=1}^{T} \text{GFT}_t(p_L^\star) + \sum_{t=1}^{T} \text{GFT}_t(p_R^\star) \geq \frac{1}{4} - \delta.$$

At this point, fix the randomness of the auxiliary sequences and focus on the one given by the coin tosses, and call $X_t$ the indicator random variable of observing left from the coin at time $t$. We have:

$$\mathbb{E}\left[\max_{p \in [0,1]} \sum_{t=1}^{T} \text{GFT}_t(p)\right] = \mathbb{E}\left[\max_{p \in \{p_L^\star, p_R^\star\}} \sum_{t=1}^{T} \text{GFT}_t(p)\right]$$

$$\geq \left(\frac{1}{4} - \delta\right) \mathbb{E}\left[\max\left\{\sum_{t=1}^{T} \mathbb{I}\{p_L^\star \in [s_t, b_t]\}, \sum_{t=1}^{T} \mathbb{I}\{p_R^\star \in [s_t, b_t]\}\right\}\right]$$

$$= \left(\frac{1}{4} - \delta\right) \mathbb{E}\left[\max\left\{\sum_{t=1}^{T} X_t, T - \sum_{t=1}^{T} X_t\right\}\right]$$

$$= \left(\frac{1}{4} - \delta\right) \mathbb{E}\left[\frac{T}{2} + \frac{1}{2}\max\left\{2\sum_{t=1}^{T} X_t - T, T - 2\sum_{t=1}^{T} X_t\right\}\right]$$

$$= \left(\frac{1}{4} - \delta\right)\left(\frac{T}{2} + \frac{1}{2}\mathbb{E}[|S_T|]\right) \geq \left(\frac{1}{4} - \delta\right)\left(\frac{T}{2} + \frac{\sqrt{T}}{3}\right),$$

where in the last inequality we used Lemma 3. Since the previous bound holds for any realization of the auxiliary sequences, it holds also in expectation over all the randomness. We can finally combine

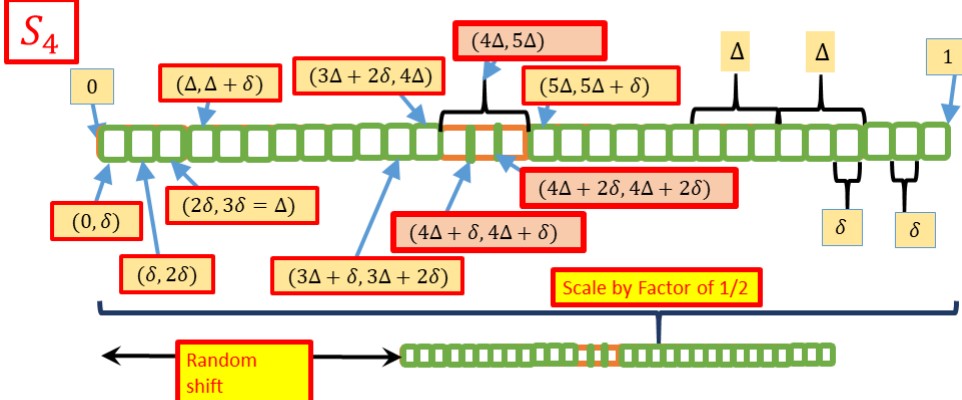

Figure 3: Example of sets used in the lower bound of Theorem 4 and how the grid is hidden. This example has $\Delta = 1/10$, $\delta = 1/30$, so each section of size $\Delta$ is partitioned into 3 sections of size $\delta$. The $(sell, buy)$ pairs in $S_4$ are as described in Equation 3 (not all such pairs are shown, there are $1/\delta = 30$ such pairs in $S_4$). Note that the gain from trade is $\Delta$ if the bids are $(4\Delta, 5\Delta)$ and if a price in between is posted. Also note that seller and buyer valuations are equal for $(4\Delta + \delta, 4\Delta + \delta)$ and for $(4\Delta + 2\delta, 4\Delta + 2\delta)$.

the two results and conclude by Yao's Theorem that

$$R_T^{2,\star} \geq \mathbb{E}\left[\max_{p \in [0,1]} \sum_{t=1}^{T} \mathrm{GFT}_t(p) - 2\sum_{t=1}^{T} \mathrm{GFT}_t(p_t)\right]$$

$$\geq \left(\frac{1}{4} - \delta\right)\left(\frac{T}{2} + \frac{\sqrt{T}}{3}\right) - \frac{T}{2}\left(\frac{1}{4} + \delta\right)$$

$$\geq \frac{1}{12}\sqrt{T} - \delta T - \delta\frac{\sqrt{T}}{3} \geq \frac{1}{13}\sqrt{T}$$

where in the last inequality we took $\delta$ small enough, e.g., $\delta = 1/T$. □

## D  Missing Proofs from Section 4

**Theorem 4** (Lower bound on $\alpha$-regret posting single price, two-bit feedback). *In the two-bit feedback model where the learner is allowed to post one single price, for all horizons $T \in \mathbb{N}$ and any constant $\alpha > 1$, the minimax $\alpha$-regret satisfies $R_T^{\alpha,\star} \geq \frac{1}{128\alpha^3}T$.*

*Proof.* In this proof, we construct a randomized family of sequences that are impossible to distinguish using a single price and given two-bit feedback. Furthermore, no deterministic algorithm is capable of achieving good regret against them in expectation. It may be useful to refer to Figure 3 for visualization. We first prove the claim under a *"grid hiding"* assumption that the learning algorithm is disallowed from posting prices in some fixed finite grid (to be defined below). We later justify the grid hiding assumption by introducing some minor perturbation to the grid.

Let $\delta$ and $\Delta$ two positive constants, with $1 \geq \Delta > \delta$ to set later such that $1/\Delta$, $1/\delta$ and $\Delta/\delta$ are integers. The grid used in the grid hiding assumption is composed of all integral multiples of $\delta$. For each $i$ from 0 to $1/\Delta - 1$, consider the following sets of valuations:

$$S_i = \left\{\left(i \cdot \Delta, (i+1)\Delta\right)\right\}$$

$$\bigcup_{j \neq i} \bigcup_{k=0}^{\Delta/\delta-1}\left\{\left(j \cdot \Delta + k \cdot \delta, j \cdot \Delta + (k+1)\delta\right)\right\} \bigcup_{k=1}^{\Delta/\delta-1}\left\{\left(i \cdot \Delta + k \cdot \delta, i \cdot \Delta + k \cdot \delta\right)\right\} \quad (3)$$

The adversary constructs the first family of sequences as follows: to start, it selects uniformly at random $i$ from 0 to $1/\Delta - 1$, then generates the sequence by repeatedly drawing independently and

uniformly at random $(s_t, b_t)$ from $S_i$. Note that the cardinality of $S_i$ is $N_\delta := 1/\delta$ for all $i$. As a first step, we give a lower bound on the expected gain from trade of the best fixed price in hindsight: fix any realization of the random draws from $S_i$ and any price $p_i^\star$ in $(i \cdot \Delta, (i+1) \cdot \Delta)$. We have then that

$$\mathbb{E}\left[\max_{p \in [0,1]} \sum_{t=1}^T \mathrm{GFT}_t(p)\right] \geq \mathbb{E}\left[\sum_{t=1}^T \mathrm{GFT}_t(p_i^\star)\right] = \frac{\Delta}{N_\delta} T = T\delta\Delta. \tag{4}$$

Since Equation (4) holds for any realization of the initial choice of $S_i$, it also holds in expectation over all the randomness of the adversary, i.e. also over the random choice of $S_i$.

Let us focus now on the expected performance of any deterministic learner $\mathcal{A}$. The crux of this proof is that any price that is not a multiple of $\delta$ is not able to discriminate between $S_i$ and $S_j$, for any $j \neq i$. To see this, let $p$ be any price that is not a multiple of $\delta$, there exist unique $j(p) \in \{0, 1, \ldots 1/\Delta - 1\}$ and $k(p) \in \{0, 1, \ldots \Delta/\delta - 1\}$ such that

$$p \in \Big(j(p) \cdot \Delta + k(p) \cdot \delta \ , \ j(p) \cdot \Delta + (k(p) + 1) \cdot \delta\Big).$$

The crucial observation is now that *regardless of the $S_i$ selected by the adversary*, the random variable $(\mathbb{P}(s_t \leq p), \mathbb{P}(p \leq b_t))$ follows the same distribution, in particular, we get:

$$(\mathbb{P}(S_t \leq p), \mathbb{P}(p \leq B_t)) = \begin{cases} (1,1) & \text{with probability } \frac{1}{N_\delta} \\ (1,0) & \text{with probability } \frac{1}{N_\delta}\left(j(p)\frac{\Delta}{\delta} + k(p) - 1\right) \\ (0,1) & \text{with the remaining probability} \end{cases}$$

Stated differently, the learner observes a trade with a fixed probability $1/N_\delta$, while the probability masses on the left and on the right are determined by the position of $p$, and are constant across all choices of $S_i$ by the adversary. Any trade that the learner observes corresponds to a $\Delta$ gain with probability $\Delta$ and to a $\delta$ gain with the remaining probability. All in all, the gain from trade for any price posted by the learner (in expectation over both the random choice of $S_i$ and the randomness at that specific round) is:

$$\mathbb{E}\left[\mathrm{GFT}_t(p_t)\right] = \frac{1}{N_\delta}\left[\frac{\Delta}{N_\Delta} + \delta\left(1 - \frac{1}{N_\Delta}\right)\right] = \delta(\Delta^2 + \delta - \Delta\delta) \leq \delta(\Delta^2 + \delta),$$

where $N_\Delta := 1/\Delta$ and represents the number of possible $S_i$ that the adversary randomly select at the beginning. Summing up the inequality over all times $t$, we get (for all learners that do not post multiples of $\delta$)

$$\mathbb{E}\left[\sum_{t=1}^T \mathrm{GFT}_t(p_t)\right] \leq \delta(\Delta^2 + \delta)T \tag{5}$$

Via Yao's Theorem and combining Equation (4) and Equation (5), we get:

$$R_T^{\alpha,\star} \geq \mathbb{E}\left[\max_{p \in [0,1]} \sum_{t=1}^T \mathrm{GFT}_t(p) - \alpha \sum_{t=1}^T \mathrm{GFT}_t(p_t)\right] \geq (\Delta - \alpha(\Delta^2 + \delta))\delta \cdot T$$

At this point we set[6] $\Delta = 1/(2\alpha)$ and $\delta = 1/(8\alpha^2)$:

$$R_T^{\alpha,\star} \geq \mathbb{E}\left[\max_{p \in [0,1]} \sum_{t=1}^T \mathrm{GFT}_t(p) - \alpha \sum_{t=1}^T \mathrm{GFT}_t(p_t)\right] \geq \frac{1}{64\alpha^3}T.$$

The proof above requires the "grid hiding" assumption that the learning algorithm cannot post prices that are on the grid (multiples of $\delta$).

One way to proceed is to scale down the instance by a constant factor, say $1/2$, so that all prices and valuations will be in the range from $0$ to $1/2$. Then the adversary adds a random uniform number (called a shift) between $0$ up to $1/2$ (see Figure 3). It is clear that any algorithm has zero probability to pinpoint the exact value of the shift, thus the learning algorithm can post a multiple of $\delta/2$ plus the required shift with probability $0$. In this scaled-down instance, the total gain from trade derived from the optimal fixed price goes down by a factor of $1/2$, while the gain of the learning algorithm is scaled down by a further factor of at least $2$ (since the learner has to deal with the extra uncertainty due to the random shift). Ergo, the $\alpha$-regret will be at least $\frac{1}{128\alpha^3}T$ which completes the proof. $\quad\square$

---

[6]At the beginning of the proof we assumed $1/\Delta$, $1/\delta$, and $\Delta/\delta$ to be integer. It is easy to see that this is without loss of generality, given these choices of $\delta$ and $\Delta$

Consider any instantiation of the algorithm BLOCK-DECOMPOSITION, fix any block $B_\tau$ and price $p$. With a slight abuse of notation we denote the average gain from trade posting price $p$ in $B_\tau$ as $\mathrm{GFT}_\tau$; formally,

$$\mathrm{GFT}_\tau(p) = \frac{1}{\Delta} \sum_{t \in B_\tau} \mathrm{GFT}_t(p).$$

We show that $\widehat{\mathrm{GFT}}_\tau(p)$ as defined in the pseudocode is an unbiased estimator of $\mathrm{GFT}_\tau(p)$, where the randomization is due to the random choice of the injective function $h_\tau$ and the inherent randomness in the estimator $\widehat{\mathrm{GFT}}$.

**Lemma 4.** *Fix any sequence of valuations, then the random variable $\widehat{\mathrm{GFT}}_\tau(p_i)$ is an unbiased estimator of $\mathrm{GFT}_\tau(p_i)$ for any $\tau \in \{1, 2, \ldots, S\}$ and price $p_i$ on the grid $Q$.*

*Proof.* For any fixed price $p_i$ it is clear that $h_\tau(p_i)$ is distributed uniformly at random in the time steps contained in the block $B_\tau$. Moreover, given $h_\tau$, the $\widehat{\mathrm{GFT}}$ are still unbiased estimators of the corresponding time steps. Thus, we have the following:

$$\mathbb{E}\left[\widehat{\mathrm{GFT}}_\tau(p_i)\right] = \sum_{t \in B_\tau} \mathbb{P}\left(h_\tau(p_i) = t\right) \mathbb{E}\left[\widehat{\mathrm{GFT}}_t(p_i) \mid h_\tau(p_i) = t\right]$$

$$= \sum_{t=1}^{T} \frac{1}{\Delta} \mathbb{E}\left[\widehat{\mathrm{GFT}}_t(p_i) \mid h_\tau(p_i) = t\right]$$

$$= \sum_{t=1}^{T} \frac{1}{\Delta} \mathrm{GFT}_t(p_i) = \mathrm{GFT}_\tau(p_i).$$

A notational observation: with the random variable $\widehat{\mathrm{GFT}}_t(p)$ we refer to the result of the estimation procedure in $p$ run at time $t$, which is an unbiased estimator of the gain from trade of price $p$ achievable at time $t$. $\qquad\square$

**Theorem 5** (Upper bound on 2-regret posting two prices, one-bit feedback)**.** *In the one-bit feedback model where the learner is allowed to post two prices, the 2-regret of BLOCK-DECOMPOSITION (BD) is such that $R_T^2(\mathrm{BD}) \leq 5T^{3/4}\sqrt{\log(T)}$, for appropriate choices of the expert algorithm $\mathcal{E}$, grid $Q$ and number of blocks $S$.*

*Proof.* We consider a grid $Q$ of equally spaced prices (we set the step later) and denote with $\Delta = T/S$ the length of every time block. The learner keeps playing the same price in each block, apart from the explorations steps, which are drawn uniformly at random. The learner decides which action to play according to some routine $\mathcal{E}$ that is run on $S$ time steps and $|Q|$ actions: this is the reason we talk interchangeably of actions and prices.

From Lemma 4 we know that the estimators in a block, i.e., $\widehat{\mathrm{GFT}}_\tau(p_i)$ are indeed unbiased estimators of $\mathrm{GFT}_\tau(p_i)$. Since this holds for any price $p_i \in Q$, it also holds for any random price $\hat{p}$ whose randomness is independent of the choice of the injection $h_\tau$ and the internal randomization of the estimators. Thus, the same holds even if instead of a fixed price $p_i$ we consider price $p_\tau$ posted by the algorithm because it depends *only* on what happened in past blocks. Let now $\mathcal{E}$ be the Multiplicative Weights algorithm. If we fix the randomness in the exploration and in the estimation upfront and consider only the inherent randomness in $\mathcal{E}$ we inherit the bound on the regret of $\mathcal{E}$ on the realized estimated gain from trades (note that they are all bounded in $[0, 1]$)

$$\max_{p \in Q} \sum_{\tau=1}^{S} \widehat{\mathrm{GFT}}_\tau(p) - \mathbb{E}\left[\sum_{\tau=1}^{S} \widehat{\mathrm{GFT}}_\tau(p_\tau)\right] \leq 2\sqrt{S \log(|Q|)} \qquad (6)$$

The randomness of $\mathcal{E}$ depends somehow on the realizations of the random injections and estimators, but if we look at any block $B_\tau$, we see that the random price output by the routine is independent of $h_\tau$ and the estimators in that block. Therefore, we can safely take the expected value (on the randomness of the $h_\tau$ and the estimators) on both sides of Equation (6), apply Lemma 4, and get

$$\max_{p \in Q} \sum_{\tau=1}^{S} \mathrm{GFT}_\tau(p) - \mathbb{E}\left[\sum_{\tau=1}^{S} \mathrm{GFT}_\tau(p_\tau)\right] \leq 2\sqrt{S \log(|Q|)}. \qquad (7)$$

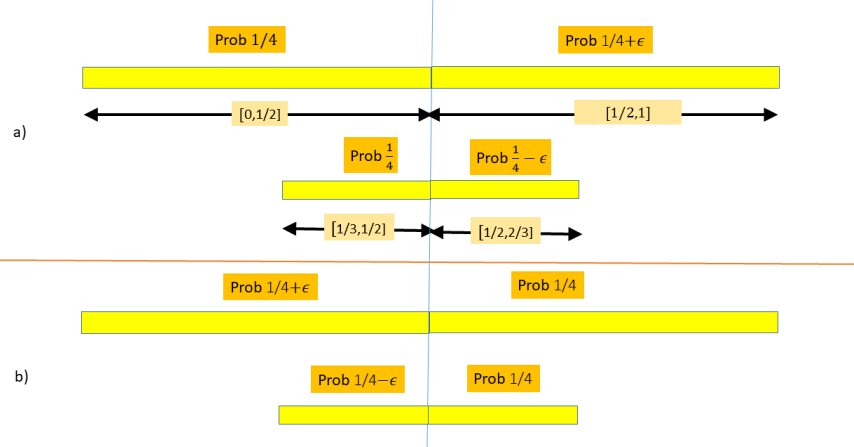

Figure 4: Construction used in the lower bounds of Theorem 6. The adversary chooses either option a) or option b). To distinguish between the two cases the algorithm must set prices in the (suboptimal) ranges $[0, 1/3)$ or $(2/3, 1]$. The expected gain from trade for a price in these segments is about $1/8$, smaller by an additive term, approximately equal to $1/6$, from the expected gain from trade achieved by placing a price somewhere in the range $[1/3, 2/3] \setminus \{1/2\}$. The optimal price is $1/2$ for the examples as described above in this figure — as done previously — $1/2$ can be slightly perturbed and thus cannot be found by the online learner. By choosing the [perturbed] price (about 1/2) the optimal gain from trade is about twice the expected gain from trade achieved by any other price.

Note that we have derived the first inequality using that the max of the expectation is smaller than the expectation of the max. Now, we move from the blocks time scale to the normal one and multiply everything by a factor $\Delta$. Our algorithm does not always play $p_\tau$, but for each one of the blocks, it spends $|Q|$ steps exploring. Therefore, we need to consider an extra $|Q|S$ additive term. At this point, we have all the ingredients to bound the 2-regret of our algorithm. Plugging Equation (7) and the observation about the extra losses incurred by the exploration into the discretization inequality (Claim 1) we get:

$$R_T^2(\mathrm{BD}) \leq 2\Delta\sqrt{S \log |Q|} + |Q|S + \delta(Q)T.$$

The theorem then follows by optimizing the free parameters: we set $\Delta = \sqrt{T}$ and we choose $Q$ to be the uniform grid of multiples of $T^{-1/4}$ (thus $S = \sqrt{T}$, $\delta(Q) = T^{-1/4}$ and $|Q| = T^{1/4} + 1$). $\quad \square$

### D.1 Lower bound on 2-regret, posting two prices and two-bit feedback

Recall the construction of the hard randomized instance for this problem: the adversary selects uniformly at random one of these two distributions at the beginning and draws $T$ i.i.d. samples from it. We report here the distributions for completeness, see Figure 4 for a pictorial representation.

$$\begin{cases} (0, \frac{1}{2}) & \text{with probability } \frac{1}{4} + \varepsilon \\ (\frac{1}{3}, \frac{1}{2}) & \text{with probability } \frac{1}{4} - \varepsilon \\ (\frac{1}{2}, \frac{2}{3}) & \text{with probability } \frac{1}{4} \\ (\frac{1}{2}, 1) & \text{with probability } \frac{1}{4} \end{cases} \qquad \begin{cases} (0, \frac{1}{2}) & \text{with probability } \frac{1}{4} \\ (\frac{1}{3}, \frac{1}{2}) & \text{with probability } \frac{1}{4} \\ (\frac{1}{2}, \frac{2}{3}) & \text{with probability } \frac{1}{4} - \varepsilon \\ (\frac{1}{2}, 1) & \text{with probability } \frac{1}{4} + \varepsilon \end{cases}$$

As we mentioned in the main text, the expected gain from trade achievable by the learner against these two distributions are:

$$\begin{cases} \frac{1}{8} + \frac{\varepsilon}{2} & \text{if } p \in [0, \frac{1}{3}) \\ \frac{1}{6} + \frac{\varepsilon}{3} & \text{if } p \in [\frac{1}{3}, \frac{1}{2}) \\ \frac{1}{3} + \frac{\varepsilon}{3} & \text{if } p = \frac{1}{2} \\ \frac{1}{6} & \text{if } p \in (\frac{1}{2}, \frac{2}{3}] \\ \frac{1}{8} & \text{if } p \in (\frac{2}{3}, 1] \end{cases} \qquad \begin{cases} \frac{1}{8} & \text{if } p \in [0, \frac{1}{3}) \\ \frac{1}{6} & \text{if } p \in [\frac{1}{3}, \frac{1}{2}) \\ \frac{1}{3} + \frac{\varepsilon}{3} & \text{if } p = \frac{1}{2} \\ \frac{1}{6} + \frac{\varepsilon}{3} & \text{if } p \in (\frac{1}{2}, \frac{2}{3}] \\ \frac{1}{8} + \frac{\varepsilon}{2} & \text{if } p \in (\frac{2}{3}, 1] \end{cases}$$

It is clear that the best price is $\frac{1}{2}$, which yields an expected gain from trade that is approximately a multiplicative factor 2 larger than the one induced by the second best price, i.e. $p \in [\frac{1}{3}, \frac{1}{2})$ or

$p \in (\frac12, \frac23]$ depending on the instance in question. The two candidates to be the second best price are an additive $\Theta(\varepsilon)$ factor away while posting prices in $[0, \frac13) \cup (\frac23, 1]$ gives a constant loss. The crucial property is that the only way the learner can discriminate between the two instances is to post prices in the low gain region $[0, \frac13) \cup (\frac23, 1]$. For example, posting a price of $\frac13$ the learner observes a trade with probability exactly $\frac12$ in both the distributions, while posting 0 yields a trade with probability $\frac14 + \varepsilon$ in the first case while exactly $\frac14$ in the second (thus allowing some learning to happen). Moreover, the learner cannot take advantage of the possibility of posting more than one price: the only useful thing to learn is where the extra $\varepsilon$ probability is, and there is no way of doing it without suffering a constant instantaneous regret; not even with two-bits of feedback. We formalize these considerations in the following lemma.

**Lemma 2.** *Consider the class of learning algorithms that can post two prices (both different from $1/2$) and receive two-bit feedback. For any $\mathcal{A}$ in this class, there exists a sequence from the family we described such that the following bound on the regret holds, for some constant $c > 0$:*

$$\max_{p \neq \frac12} \sum_{t=1}^{T} \mathrm{GFT}(p) - \mathbb{E}\left[ \sum_{t=1}^{T} \mathrm{GFT}_t(p_t, q_t) \right] \geq cT^{2/3}.$$

To prove the Lemma, we show that the family of sequences presented above fits into the proof scheme of Theorem 4 of Bartók et al. [2014]. To formally show this, we need to introduce the theoretical framework of partial monitoring and argue that our instance is a special case of the one used in the $\Omega(T^{2/3})$ lower bound of Theorem 4 of Bartók et al. [2014].

We recall from Bartók et al. [2014] that an $N$ actions, $M$ outcomes partial monitoring game is characterized by two matrices, the loss (or gain, as in our case) matrix $L$ and the signal matrix $H$. In each round $t$, the learner chooses an action $I_t \in [N]$ and, simultaneously, the adversary chooses an outcome $J_t \in [M]$. The learner experiences a gain $L_{I_T, J_T}$ and receives as feedback $H_{I_T, J_T}$. The notion of regret is defined as in the classical online learning framework as the difference between the total gain of the best fixed action in hindsight and the expected gain of the learning algorithm.

| | $(0,\frac12)$ | $(\frac13,\frac12)$ | $(\frac12,\frac23)$ | $(\frac12,1)$ |
|---|---|---|---|---|
| $(0,0)$ | $\frac12$ | 0 | 0 | 0 |
| $(0,\frac13)$ | $\frac12$ | 0 | 0 | 0 |
| $(0,\frac23)$ | 0 | 0 | 0 | 0 |
| $(0,1)$ | 0 | 0 | 0 | 0 |
| $(\frac13,\frac13)$ | $\frac12$ | $\frac16$ | 0 | 0 |
| $(\frac13,\frac23)$ | 0 | 0 | 0 | 0 |
| $(\frac13,1)$ | 0 | 0 | 0 | 0 |
| $(\frac23,\frac23)$ | 0 | 0 | $\frac16$ | $\frac12$ |
| $(\frac23,1)$ | 0 | 0 | 0 | $\frac12$ |
| $(1,1)$ | 0 | 0 | 0 | $\frac12$ |

Table 2: Gain Matrix $L$

| | $(0,\frac12)$ | $(\frac13,\frac12)$ | $(\frac12,\frac23)$ | $(\frac12,1)$ |
|---|---|---|---|---|
| $(0,0)$ | $(1,1)$ | $(0,1)$ | $(0,1)$ | $(0,1)$ |
| $(0,\frac13)$ | $(1,1)$ | $(0,1)$ | $(0,1)$ | $(0,1)$ |
| $(0,\frac23)$ | $(1,0)$ | $(0,0)$ | $(0,1)$ | $(0,1)$ |
| $(0,1)$ | $(1,0)$ | $(0,0)$ | $(0,0)$ | $(0,1)$ |
| $(\frac13,\frac13)$ | $(1,1)$ | $(1,1)$ | $(0,1)$ | $(0,1)$ |
| $(\frac13,\frac23)$ | $(1,0)$ | $(1,0)$ | $(0,1)$ | $(0,1)$ |
| $(\frac13,1)$ | $(1,0)$ | $(1,0)$ | $(0,0)$ | $(0,1)$ |
| $(\frac23,\frac23)$ | $(1,0)$ | $(1,0)$ | $(1,1)$ | $(1,1)$ |
| $(\frac23,1)$ | $(1,0)$ | $(1,0)$ | $(1,0)$ | $(1,1)$ |
| $(1,1)$ | $(1,0)$ | $(1,0)$ | $(1,0)$ | $(1,1)$ |

Table 3: Feedback Matrix $H$

The two tables represent the gain and feedback matrices of the family of sequences we introduced in the main body. Note that the rows refer to the actions, i.e., the prices posted to seller and buyers, while the columns to the outcomes, *i.e.*, the valuations of the agents. The row-colors reflect the properties of the corresponding actions: green for Pareto-Optimal, yellow for degenerate and white for dominated.

In our family of instances, we have $M = 4$ possible outcomes, according to the valuations: $(s, b) = (0, \frac12), (\frac13, \frac12), (\frac12, \frac23)$ or $(\frac23, 1)$. The possible actions correspond to all the possible (continuous) prices posted by the learner; however, given the structure of the problem, it is enough to consider only a finite representative set of $N = 10$ of them: $(q, p) \in \{0, \frac13, \frac23, 1\}^2$ such that $q \leq p$. Note that this

| | $(0, \frac{1}{2})$ | $(\frac{1}{3}, \frac{1}{2})$ | $(\frac{1}{2}, \frac{2}{3})$ | $(\frac{1}{2}, 1)$ |
|---|---|---|---|---|
| $(1,1)$ | 1 | 0 | 0 | 0 |
| $(1,0)$ | 0 | 0 | 0 | 0 |
| $(0,1)$ | 0 | 1 | 1 | 1 |
| $(0,0)$ | 0 | 0 | 0 | 0 |

Table 4: Signal matrices of actions 1 and 2

| | $(0, \frac{1}{2})$ | $(\frac{1}{3}, \frac{1}{2})$ | $(\frac{1}{2}, \frac{2}{3})$ | $(\frac{1}{2}, 1)$ |
|---|---|---|---|---|
| $(1,1)$ | 1 | 1 | 0 | 0 |
| $(1,0)$ | 0 | 0 | 0 | 0 |
| $(0,1)$ | 0 | 0 | 1 | 1 |
| $(0,0)$ | 0 | 0 | 0 | 0 |

Table 5: Signal matrix of action 5

| | $(0, \frac{1}{2})$ | $(\frac{1}{3}, \frac{1}{2})$ | $(\frac{1}{2}, \frac{2}{3})$ | $(\frac{1}{2}, 1)$ |
|---|---|---|---|---|
| $(1,1)$ | 0 | 0 | 0 | 0 |
| $(1,0)$ | 1 | 1 | 0 | 0 |
| $(0,1)$ | 0 | 0 | 0 | 0 |
| $(0,0)$ | 0 | 0 | 1 | 1 |

Table 6: Signal matrix of action 8

| | $(0, \frac{1}{2})$ | $(\frac{1}{3}, \frac{1}{2})$ | $(\frac{1}{2}, \frac{2}{3})$ | $(\frac{1}{2}, 1)$ |
|---|---|---|---|---|
| $(1,1)$ | 0 | 0 | 0 | 0 |
| $(1,0)$ | 1 | 1 | 1 | 0 |
| $(0,1)$ | 0 | 0 | 0 | 0 |
| $(0,0)$ | 0 | 0 | 0 | 1 |

Table 7: Signal matrices of actions 9 and 10

is without loss of generality since prices in the same interval gets the same gain and feedback. To get more intuition: the action *posting prices* $(0, 1/3)$ represents all the actions where the price to the seller is in the $[0, \frac{1}{3})$ interval and the price to the buyer is in the $[1/3, 1/2)$ interval. The gain and feedback matrices are reported in Tables 2 and 3.

Let $\Delta_M$ denote the $M$-dimensional probability simplex, and let $\ell_i$ denote the $i^{th}$ row of $L$ as a column array. Given a vector $\pi \in \Delta_M$, this induces a probability distribution over the outcomes; an action $i$ is optimal under $\pi$ if it is the best response of the learner to $\pi$: $\langle \ell_i, \pi \rangle \geq \langle \ell_{i'}, \pi \rangle$ for all $i' \neq i$. The notion of optimal action induces a cell decomposition of $\Delta_M$.

**Definition 1** (Cell Decomposition). *For every action $i \in [N]$, let*

$$C_i = \{\pi \in \Delta_M : \text{ action } i \text{ is optimal under } \pi\}.$$

*The sets $C_1, \ldots, C_N$ constitute the cell decomposition of $\Delta_M$.*

As an example, consider the first action in Table 2, corresponding to posting price 0 to both agents. Its cell $C_1$ in $\Delta_4$ is composed by all $\pi = (\pi_1, \pi_2, \pi_3, \pi_4) \in \Delta_4$ such that action 1 is the best response to it. Clearly, $\pi_2 = 0$, because otherwise the fifth action $(\frac{1}{3}, \frac{1}{3})$ would guarantee strictly larger gain from trade. The last two actions give the only other constraint:

$$\frac{1}{2}\pi_1 \geq \frac{1}{6}\pi_3 + \frac{1}{2}\pi_4.$$

Therefore $C_1 = \{\pi \in \Delta_4 : \pi_2 = 0, \ 3\pi_1 - \pi_3 - 3\pi_4 \geq 0\}$. Using the cell decomposition, we can characterize the actions.

- Action $i$ is called dominated if $C_i = \emptyset$, otherwise it is called non-dominated. In our instance, actions $3, 4, 6$ and $7$ are dominated, and the others are non-dominated.

- Action $i$ is called degenerate if it is non-dominated and there exists an action $i'$ such that $C_i \subsetneq C_{i'}$. Our first two actions and the last two are degenerate.

- If an action is neither dominated nor degenerate, it is called Pareto-optimal. In our instance, actions 5 and 8 are Pareto-optimal.

- Two Pareto-optimal actions $i$ and $j$ are neighbors if $C_i \cap C_j$ is an $(M - 2)$-dimensional polytope. The neighborhood action set of two neighboring actions $i$ and $j$ is defined as $N_{i,j}^+ = \{k \in [N] : C_i \cap C_j \subseteq C_k\}$. In our example $C_5 \cap C_8 = \{\pi \in \Delta_4 : 3\pi_1 + \pi_2 = \pi_3 + 3\pi_4\}$. Therefore the two Pareto-optimal actions are neighbors whose neighborhood contains only the two of them.

We now move our attention to the feedback matrix.

**Definition 2.** *Let $s_i$ be the number of symbols in the $i^{th}$ row of $H$ and let $\sigma_1, \ldots, \sigma_{s_i}$ be an enumeration of those symbols. Then the signal matrix $S_i \in \{0,1\}^{s_i \times M}$ of action $i$ is defined as $(S_i)_{k,\ell} = \delta_{H_{i,\ell} = \sigma_k}$.*

To get a better understanding of the definition, note that in our example, the symbols are $4$ and correspond to the possible two-bit feedback: $(1,1)$, $(1,0)$, $(0,1)$, and $(0,0)$. The signals matrices of the non-dominated actions are reported in Tables 4, 5, 6 and 7 (for the sake of uniformity we reported all the symbols in each signal matrix, this does not affect the results in any way).

We are ready to introduce the key definitions of observability we need to invoke the characterization theorem.

**Definition 3.** *A partial monitoring game admits the global observability condition if, for all pairs $i$ and $j$ of actions, the vector $\ell_i - \ell_j$ belongs to the span generated by all the rows of the signal matrices:*

$$\ell_i - \ell_j \in \bigoplus_{k \in [N]} Im(S_k^T).$$

This definition seems fairly abstract, but in our case is extremely easy to verify: the first row of $S_1$ (Table 4), the first and third rows of $S_5$ (Table 5) and the last row of $S_9$ (Table 7) generate all $\mathbb{R}^4$. Hence, our game respects the global observability condition.

**Definition 4.** *A pair of neighboring actions $i, j$ is said to be locally observable if*

$$\ell_i - \ell_j \in \bigoplus_{k \in N_{i,j}^+} Im(S_k^T).$$

*A game satisfies the local observability condition if every pair of neighboring actions is locally observable.*

Our instance does not respect the local observability condition. We already argued that $N_{5,8}^+$ contains only the two actions $5$ and $8$. If we look at the span generated by the row vectors of their signal matrices, we observe that it consists of all the vectors $v = (v_1, v_2, v_3, v_4) \in \mathbb{R}^4$ that can be written as $(\lambda, \lambda, \mu, \mu)$ for some parameters $\lambda$ and $\mu$, while $\ell_5 - \ell_8 = \left(\frac{1}{2}, \frac{1}{6}, -\frac{1}{6}, -\frac{1}{2}\right)$. So far, we have shown enough to claim that the instance we have built is an "hard" partial monitoring game [Bartók et al., 2014].

*Proof of Lemma 2.* Theorem 4 of Bartók et al. [2014] states that the minimax regret of a partial monitoring instance that does not respect the local observability condition is at least $c \cdot T^{2/3}$, for some instance-specific constant $c$. To conclude the proof of the Lemma, we note that the probability distributions over the instances we stated in the previous section are indeed a special case of the ones used in the main body of Theorem 4 of Bartók et al. [2014]. More in the specific, actions $1$ and $2$ in that proof correspond to our actions $5$ and $8$, probability vector $\left(\frac{1}{4}, \frac{1}{4}, \frac{1}{4}, \frac{1}{4}\right)$ corresponds to what is there denoted with $p_0$ (note that that this choice of $p_0$ does belong in ours $C_5$ and $C_8$) and our choice of vector $v$ is $v = (1, -1, 0, 0)$ (up to a rescaling of the $\varepsilon$ small enough). $\qquad \square$