# OpenReview forum: "An $\alpha$-regret analysis of Adversarial Bilateral Trade"
_NeurIPS.cc/2022/Conference — NeurIPS 2022 Accept_

### Official Review · Reviewer_cUb6 · 2022-07-06

**Rating:** 7
**Confidence:** 3
**Soundness:** 3 good
**Presentation:** 3 good
**Contribution:** 3 good

**Summary:**

This paper studies online bilateral trade in the adversarial setting. Previously, there had been some work on online bilateral trade but in the stochastic setting. The authors prove a number of results. First, note that one cannot hope to have an online algorithm that is able to compete with the optimal GFT (this is a result due to Cesa-Bianchi et al.).They further strengthen this to show that one cannot obtain any approximation ratio better than 1/2 even with full feedback, even in the adversarial setting.

On the upper bound side, the authors show that sqrt(T log T) 2-regret is possible when full feedback is available.

They also consider different feedback models. In the one-bit feedback model, one is allowed to post a single price and observes only the event of whether or not a trade occurs. In the two-bit feedback model, one sees two events: the event that the buyer is willing to trade and the event that the seller is willing to trade. They also consider two ways of “pricing”. The first is to use a single price and the second is to use two prices.

On the upper bound side, they show that in the one-bit feedback model, can obtain 2-regret of T^{3/4} by using two prices. If one uses only a single price then one has 2-regret at least T. They also show T^{2/3} 2-regret with two prices in the two-bit feedback model.

**Questions:**

- Mention in abstract that only a single seller and single buyer is considered.
- Line 288, missing reference
- Out of curiosity, I wonder if the authors consider multiple buyers and/or multiple sellers at all?
- Line 50, it mentions that Deng et al give a BIC mechanism for approximating the first-best GFT. This is slightly inaccurate in that they analyzed a known mechanism. For example, the mechanism was also discussed in "Approximating Gains from Trade in Two-sided Markets
via Simple Mechanisms" by Brustle, Cai, Wu, and Zhao.

**Limitations:**

Limitations discussions looks good. They filled out the checklist.

**Strengths And Weaknesses:**

**Strengths.**
- I think the problem is very nice and I think the authors prove a number of very nice results. The techniques are very nice (Lemma 1 is a very neat trick). I think this paper is a pretty solid contribution.

**Weaknesses.**
- None

---

> ### Author Response · Authors · 2022-08-02
> **Our response to Reviewer cUb6**
>
> $\bf{Questions:}$
>
> - Mention in abstract that only a single seller and single buyer is considered.
>
> [Answer]: We will clarify in the abstract that we consider only a single seller and a single buyer.
>
>
> - Line 288, missing reference
>
> [Answer]: We will fix the missing reference.
>
>
> - Out of curiosity, I wonder if the authors consider multiple buyers and/or multiple sellers at all?
>
> [Answer]: We are thinking about models for multiple buyers and sellers, but there is no obvious single generalization.
>
>
> - Line 50, it mentions that Deng et al give a BIC mechanism for approximating the first-best GFT. This is slightly inaccurate in that they analyzed a known mechanism. For example, the mechanism was also discussed in "Approximating Gains from Trade in Two-sided Markets via Simple Mechanisms" by Brustle, Cai, Wu, and Zhao.
>
> [Answer]: We will cite this paper as well.

---

> > ### Comment · Reviewer_cUb6 · 2022-08-08
> > **thanks**
> >
> > Thanks for the response.

---

### Official Review · Reviewer_wuuk · 2022-07-08

**Rating:** 6
**Confidence:** 3
**Soundness:** 3 good
**Presentation:** 2 fair
**Contribution:** 2 fair

**Summary:**

This paper studies the problem optimizing gains from trade when a buyer with value b and seller with cost s arrive online. The process sets a single price p = q or two prices p and q, a trade happens if s <= p <= q <= b. The gain from trade is (b-s). Gain from trade is harder objective to optimize as the value is zero if no trade happens.

The authors consider online pricing with the goal of minimizing regret compared to the optimal gains from trade in hindsight. In this model 1-regret is not possible, so authors consider 2-regret where the optimal hindsight gains from trade - 2 x achieved gains from trade is sub-linear in the number of steps T.

They consider three different models on what information is available: Full feedback where the planner learns (s, b), 2 bit feedback where the planner learns if (s < p) and (q < b) and 1 bit feedback where the planner learns if the trade happened or not. The authors provide variety of lower bounds on the regret-approximation and regret as a function of T and provide close/matching upper bounds.

In particular the authors show.
* a lower bound of 2 for the regret approximation in full feedback model
* A matching upper and lower bound of O(\sqrt(T)) for the regret in the full feedback model.
* A lower bound of O(T) for regret with two bit feedback and single price.
* A lower bound of O(T^{2/3})  with two prices and two bit feedback.
* Upper bound of O(T^{3/4}) with two prices and 1 bit feedback.

**Questions:**

* Please clarify the details for the result in Section 4.2.

**Limitations:**

Authors note some of the gaps in their results. There are no societal impact implications to note.

**Strengths And Weaknesses:**

The results in this paper are comprehensive and the questions studied seem interesting.

The results are non-trivial. The upper bound in the full information model comes from multiplicative weights. For the two lower bound the authors provide specific constructions that have high gains from trade in hindsight but the gains from trade of any pricing algorithm are much lower. The lower bound constructions for the two bid feedback models are much more involved. The upper bound for two bit feedback model is also involved.

A big weakness is in the presentation of the paper. A lot of the details in the main body are difficult to follow without consulting the appendix. The appendix itself has also becomes difficult to follow due to having to refer back to the main paper. Perhaps the authors could find a better way to split the content. As an example, the lower bound constructions in Figure 1 and Figure 2 made no sense without the verbal description.

Another concern is about the correctness of some of the results. The more involved results in the paper are also hard to follow. In particular, Section 4.2 does not make complete sense even after putting it together with the appendix.
The estimator \hat{GFT} is defined without a subscript but is later used with a variety of subscripts. The subscript h^{-1}(p) seems like a typo and should be h(p). The subscript \tau refers to a batch of requests. It's not clear what is going on here.
I am also not sure how \hat(GFT)(p_i) is evaluated when the model only allows posting one pair of prices. Please clarify in the block decomposition algorithm, what price is posted in step 10. It will be important for authors to clarify this and assure that their result is indeed correct.

I also found some typos in the lower bound result for the two bit feedback model and haven't checked the correctness of the rest of the result as it is fairly complex. Please consider including a statement of Theorem 4 from Bartok et al. for completeness.

Other typos:
Page 17, line 629: define N_{\Delta}
Page 17, line 632: Claim only holds for integer \alpha. But the extension to other values in between follows.
Page 18, line 649: Clarify what q is.
Page 19, line 692:
Why is E[GFT(p)] for p in [0, 1/3] not 1/8? And the next line not 1/8 + 1/24?

Update after rebuttal:
The authors response about the GFT estimator’s definition makes sense to me and I no longer think that the approach us fundamentally wrong. I will look at the proof some more.

---

> ### Author Response · Authors · 2022-08-02
> **Our Response to Reviewer wuuk**
>
> $\bf{Weaknesses:}$
>
> - A big weakness is in the presentation of the paper. A lot of the details in the main body are difficult to follow without consulting the appendix. The appendix itself has also become difficult to follow due to having to refer back to the main paper. Perhaps the authors could find a better way to split the content. As an example, the lower bound constructions in Figure 1 and Figure 2 made no sense without the verbal description.
>
> [Answer]: We will make an effort to improve the presentation of the paper and attempt a better partition of the content between the main body and the appendix.
>
> - Another concern is about the correctness of some of the results. The more involved results in the paper are also hard to follow. In particular, Section 4.2 does not make complete sense even after putting it together with the appendix. The estimator \hat{GFT} is defined without a subscript but is later used with a variety of subscripts. The subscript h^{-1}(p) seems like a typo and should be h(p). The subscript \tau refers to a batch of requests. It's not clear what is going on here. I am also not sure how \hat(GFT)(p_i) is evaluated when the model only allows posting one pair of prices. Please clarify in the block decomposition algorithm, what price is posted in step 10. It will be important for authors to clarify this and assure that their result is indeed correct.
>
> [Answer]: We have indeed verified that these are only typos. That said, we are convinced to a very high degree of certainty that there is nothing wrong with the proofs and the results.
>
> In Lemma 1, we stated the result for a generic round of the process, without specifying the dependence on the time. The Lemma states that, given a fixed price $p$ and any two fixed but hidden valuations $s$ and $b$, it is possible to construct an unbiased estimator of the gain from trade achievable by posting price $p$ using two randomized prices (possibly different from $p$) and receiving the one bit feedback. The details of the estimator procedure are described in the pseudocode at the top of page 8.
>
> In Lemma 2, we use the subscript to refer to the specific time step when the estimator is used. We will clarify it in the final version. We thank the reviewer for pointing out the typo: it should indeed be h(p) instead of h^{-1}(p).
>
> We acknowledge that we overloaded the notation by using the subscript of $\widehat {GFT}$ both for the block number and for the specific time step within the block where the estimator is used, and that this might cause confusion. We will use a different notation.
>
> What we do is the following: for each block $B_{\tau}$, we extract uniformly at random one time step for each price $p$ and use that time step to estimate (via our estimator procedure described at the top of page 8) the gain from trade extracted from $p$ along the whole block $B_{\tau}$ (divided by $\Delta$, the length of $B_{\tau}$). The injective function h is used to model this idea of associating prices to uniformly random time steps. In line 10 the algorithm uses the pseudocode described at the top of page 8 to estimate the gain from trade obtainable at that time step by posting price $p_i$.
>
> - I also found some typos in the lower bound result for the two bit feedback model and haven't checked the correctness of the rest of the result as it is fairly complex. Please consider including a statement of Theorem 4 from Bartok et al. for completeness.
>
> [Answer]: We will add it.
>
> - Other typos: Page 17, line 629: define N_{\Delta} Page 17, line 632: Claim only holds for integer \alpha. But the extension to other values in between follows. Page 18, line 649: Clarify what q is. Page 19, line 692: Why is E[GFT(p)] for p in [0, 1/3] not 1/8? And the next line not 1/8 + 1/24?
>
> [Answer]: We thanks the reviewer for pointing these typos out. We will fix them. Specifically, in the display in line 692, the plus-minus should be either plus or 0 (e.g. in the first line $\frac 18 +\varepsilon/2$ for the first distribution and simply $\frac 18$ for the second one), while the minus-plus should be either minus or 0 (e.g. in the last line $\frac 18$ for the first distribution and $\frac 18-\varepsilon/2$ for the second one).
> So it is correct that $\mathbb{E}[GFT(p)] = 1/8$ for $p \in [0, 1/3]$ when the underlying distribution is the second one described in the display after line 690 and that in the following line $\mathbb{E}[GFT(p)] = 1/8 + 1/24 $ for $p \in (\frac 13, \frac 12)$ (still, when the underlying distribution is the second one). We note that this inaccuracy does not invalidate the correctness of the result as the difference between the two scenarios is still $\Theta(\varepsilon)$, and that is the relevant feature to apply Theorem 4 from Bartok et al..
>
> $\bf{Questions:}$
> - Please clarify the details for the result in Section 4.2.
>
> [Answer]: As mentioned above on Section 4.2, we will clarify the details.

---

> > ### Comment · Reviewer_wuuk · 2022-08-08
> > **Thank you for the clarification**
> >
> > Thank you for your clarification. The explanation regarding GFT makes sense. I was concerned that multiple prices were being posted but now I see that only one price is posted but it is randomized thus providing better odds of having a trade and providing an estimate for gains from trade. I will raise my score slightly, I need to take a more careful look at the proofs to make sure that they are okay.

---

### Official Review · Reviewer_5g6N · 2022-07-11

**Rating:** 6
**Confidence:** 3
**Soundness:** 3 good
**Presentation:** 2 fair
**Contribution:** 3 good

**Summary:**

This paper studies the sequential adversarial bilateral trade problem under various feedback models: 1) full feedback 2) one-price 1 bit feedback 3) two-price 2 bit feedback 4) two-price 1 bit feedback. This paper provides a regret lower bound of 2-approximation for the full feedback model. Additionally, they give a learning algorithm that achieves $\tilde{O}(\sqrt{T}) 2$ regret in this setting. Then, for the partial feedback model with 1 or 2-bit feedback, the authors shows that no learning algorithm can achieve sublinear regret for constant $\alpha$. Moreover, the paper provides a separation result between partial feedback and full feedback. For all the above results, the benchmark solution is the best fixed price in hindsight.

**Questions:**

1. Is there any \emph{existing} regret lower bound on adversarial bilateral trade similar to the settings of this paper?

2. Does the gain from trade results of this paper implies improvement for the social welfare benchmark?

3. Is the two-price setting realizable?

4. What did the paper mean by sublinear $\alpha$-regret when $\alpha > 1$. Conventionally, sublinear regret means the total regret is sublinear in total time horizon $T$, hence algorithm with sublinear regret can be viewed as near-optimal.


**Limitations:**

If the paper could explain how the "gain from trade" benchmark outperforms the standard social welfare benchmark, that would be great.


**Strengths And Weaknesses:**

Strength:

1. Section 1.2 introduces in detail the technical challenges of this paper, as well as related literature.

2. This paper provides intensive theoretical analysis on the proposed algorithm in various settings.

3. This paper provides a regret lower bound on full feedback model.

Weakness:
1. Some of the notations used in this paper are not standard, e.g., sublinear $\alpha$-regret.

2. The result comparison table (Table 1) is informal. The paper should specify which is the lower bound for the specific setting, and which is the performance of the proposed algorithm, by adding more descriptions to the table.

3. The review find it hard to follows some proof intuitions, even with figures (Figure 1 and Figure 2)

4. This paper doesn't have runtime analysis or experiments.

Minor Comments:
1. [line 86] the full feedback model achieves $\tilde{O}(\sqrt{T}) 2$-regret. What does it mean?

2. [line 288] broken link

---

> ### Author Response · Authors · 2022-08-02
> **Our Response to Reviewer 5g6N - The notion of $\alpha$-regret**
>
> We start our response by addressing a general concern raised by the reviewer on multiple occasions.
>
> In this paper we study the notion of $\alpha$-regret that has been formally introduced in Kakade et al (2009) and was already present, to some extent, in the seminal work of Kalai and Vampala (2005). $\alpha$-regret generalizes the concept of regret in online learning to scenarios where achieving sublinear regret (i.e. sublinear 1-regret) is impossible and has been studied extensively (see lines 152-155 in the related work Section).
> We give the relevant definition in the introduction (line 68) as well as in the last two paragraphs of the preliminaries before Section 2.1. Our results show that near optimal algorithms are impossible and a multiplicative factor 2 is required and sufficient.

---

> ### Author Response · Authors · 2022-08-02
> **Our Response to Reviewer 5g6N**
>
> $\bf{Weaknesses:}$
>
> 1. Some of the notations used in this paper are not standard, e.g., sublinear alpha-regret.
>
> [Answer]: As mentioned in the first comment, alpha-regret has been studied and defined before in various papers.
>
> 2. The result comparison table (Table 1) is informal. The paper should specify which is the lower bound for the specific setting, and which is the performance of the proposed algorithm, by adding more descriptions to the table.
>
> [Answer]: We thank the reviewer for this comment and we will add some clarifications to the table where required.
>
> 3. The review find it hard to follows some proof intuitions, even with figures (Figure 1 and Figure 2)
>
> [Answer]: we will make our best effort to make clarifications for the proofs.
>
> 4. This paper doesn't have runtime analysis or experiments.
>
> [Answer]: Our algorithms are easily implementable and the computation needed in each time step is always sublinear in $T$. Typically, our algorithms either need to sample from and update the probability vector of the prices considered (that can be done linearly in the dimension of the price-grid) or use the estimator described in page 8, that can be implemented in constant time. Finally, we mention that in our block decomposition algorithm, we only need to recompute the probability vector once in each block, thus substantially decreasing the amortized-per-step computation.
>
> $\textbf{Minor Comments:}$
>
> - [line 86] the full feedback model achieves $\tilde{O}(\sqrt{T})$ $2$-regret. What does it mean?
>
> [Answer]: As defined, it means that the multiplicative factor ($\alpha$) is $2$, while the additive term is $\tilde O(\sqrt{T})$.
>
> - [line 288] broken link
>
> [Answer]: We thank the reviewer for pointing this out, the link should have referred to Lemma 3 in the appendix, in the pdf of the supplementary materials the broken link does not appear.  It will be fixed.
>
> $\textbf{Questions:}$
>
> -Is there any \emph{existing} regret lower bound on adversarial bilateral trade similar to the settings of this paper?
>
> [Answer]: In Cesa Bianchi et al. (2021), the authors provide a linear lower bound on the regret for the adversarial sequential bilateral trade problem. In our work we prove lower bounds for 2-regret, that is a weaker notion than (1-)regret, and hence harder to prove.
>
> - Does the gain from trade results of this paper imply improvement for the social welfare benchmark?
>
> [Answer]: Note, as in Cesa Bianchi et al. (2021), that when we consider (1-)regret, gain from trade and social welfare are equivalent, since the difference is an additive constant that does not depend on the prices posted. When we discuss alpha-regret, for $\alpha > 1$, an upper bound for gain from trade naturally implies an upper bound on social welfare, but not vice-versa.
>
> - Is the two-price setting realizable?
>
> [Answer]: As we view it, all our algorithms are easy to implement. In particular note that the only point where we post two different prices is in the estimator procedures in Block Decomposition. From the game theoretic perspective, the idea of posting two different prices respecting budget balance in bilateral trade has been studied extensively in the literature (see Related work, in particular the paragraph starting at line 44)
>
> - What did the paper mean by sublinear $\alpha$-regret when $\alpha > 1$. Conventionally, sublinear regret means the total regret is sublinear in total time horizon, hence algorithm with sublinear regret can be viewed as near-optimal.
>
> [Answer]: We refer to our first comment about $\alpha$-regret. Our results show that near optimal algorithms are impossible and a multiplicative factor 2 is required and sufficient.
>
> $\textbf{Limitations:}$
>
> - If the paper could explain how the "gain from trade" benchmark outperforms the standard social welfare benchmark, that would be great.
>
> [Answer]: It is true that the difference between gain from trade and social welfare is just an additive constant (independent of the price posted), however, the notion of alpha-regret involves a multiplicative factor. It is well known that it is much harder to obtain a multiplicative approximation for gain from trade than from social welfare: for instance, the problem of obtaining a constant factor approximation for bayesian gain from trade was open for many years while the one for social welfare is easy (see, e.g. Deng et al. (2021), Duetting et al. (2021) and references therein).

---

> > ### Comment · Reviewer_5g6N · 2022-08-06
> > **Thanks for the author response**
> >
> > Thanks for the author's response. I really appreciate it.
> >
> > - The reviewer still finds it hard to understand some of the proof intuitions, this alone might lead to score $\leq 5$. (Reviewer wuuk also points this out. )
> > -  With respect to "Gain from trade versus Social Welfare" benchmark: The reviewer would like to know why the "gain from trade" benchmark is interesting, not from the difficulty of the problem itself. Specifically, the reviewer like to know whether current "gain from trade" result leads to an improvement on results using social welfare benchmark on the similar settings.
> >
> > Again, thanks for the detailed response.

---

> > > ### Author Response · Authors · 2022-08-07
> > > **Further comments**
> > >
> > > We thank the reviewer for the opportunity to address these points.
> > >
> > > - The reviewer still finds it hard to understand some of the proof intuitions, this alone might lead to score $\le 5$ . (Reviewer wuuk also points this out. )
> > >
> > > [Answer]: We acknowledge that the proof intuitions in Section 3 are hard to follow without the formal descriptions of the Figures provided in the appendix (as mentioned by Reviewer wuuk). We will use the extra page of the camera ready to move that part in the main text and improve the presentation. That said, we are convinced to a very high degree of certainty that there is nothing wrong with the proofs and the results. If the reviewer has other specific concerns about the presentation of our results we will be happy to provide further clarifications.
> > >
> > > - With respect to "Gain from trade versus Social Welfare" benchmark: The reviewer would like to know why the "gain from trade" benchmark is interesting, not from the difficulty of the problem itself. Specifically, the reviewer like to know whether current "gain from trade" result leads to an improvement on results using social welfare benchmark on the similar settings.
> > >
> > > [Answer]: Social welfare and gain from trade are not simply benchmarks to compete against, but two different objective functions that measure the economic efficiency of a bilateral trade mechanism. Social welfare considers the final happiness of the system, while the gain from trade measures the improvement in social welfare given by the mechanism. Informally, social welfare measures how good things are, while gain from trade measures how much things improve.
> > >
> > > Commercial platforms can gain proportionally to the gain from trade and not social welfare, this is because commercial activity (trade) is generated when there is sufficient gain from trade to motivate both sellers and buyers to join a given platform after some of the surplus has been allocated to the operation of the trading platform and for profit. In addition, gain from trade is invariant under additive shift in the values (positive and negative) while social welfare is sensitive to additive shift and defined only for positive values.
> > >
> > > Both social welfare and gain from trade have been studied extensively in the literature and share many similarities. In particular:
> > > i) if we are interested in exact maximization, then gain from trade and social welfare are equivalent
> > > ii) if we are interested in (1-)regret minimization, then the two objectives are equivalent: step-wise, gain from trade and social welfare differ by the seller’s valuation, independently of the price posted.
> > > iii) if we are interested in multiplicative approximation, then social welfare is easier to approximate (see examples below). More precisely, a $\beta$-approximation for gain from trade implies a $\beta$-approximation for social welfare; conversely, a lower bound for social welfare implies a lower bound for gain from trade. The same holds for $\alpha$-regret.
> > >
> > > As an example, imagine an instance where the seller’s valuation is $1-\delta$, while the buyer's valuation is $1$, for some small $\delta$. The social welfare of any mechanism is at least $1-\delta$, whether trade occurs or not, while the best mechanism could achieve only a very marginal (in relative terms) improvement in social welfare (i.e. make the trade happen and raise the social welfare to $1$). However, gain from trade is $0$, unless the mechanism posts a price $p$ in the narrow $[1-\delta,1]$ interval. It is hard to hit such a narrow interval.
> > >
> > > From another perspective, consider the fact that if the seller’s valuation is lower bounded by some constant $c$ and the buyer’s valuation is constrained to lie in the $[0,1]$ interval, then any mechanism would give a multiplicative 1/c-approximation to the optimal social welfare, Clearly this is not true if we look at the gain from trade: if the mechanism fails to make the trade happen, then the gain from trade is 0 and the approximation unbounded.
> > >
> > > Finally, the only paper that studies sequential bilateral trade is Cesa-Bianchi et al. (2021). There the only result for the adversarial case (that we study) is a negative one: a linear lower bound on the (1-)regret, that clearly applies to both social welfare and gain from trade. In our paper, we improve on this result for both objectives. In particular, all the positive results for gain from trade are valid for social welfare. We do believe that the study of alpha-regret is also interesting for social welfare and we already have some preliminary results in this direction (namely, the social welfare equivalent of our Theorem 1), but this is out of the scope of this current paper.

---

> > > > ### Comment · Reviewer_5g6N · 2022-08-08
> > > > **Thanks for the helpful comments**
> > > >
> > > > From the example, the reviewer really appreciates the responses during the rebuttal period and understands the importance of gain-from-trade, and how it's different from the social welfare benchmark. The reviewer has changed the score accordingly. Additionally, the reviewer suggests adding these discussions to the paper.

---

### Official Review · Reviewer_VWf6 · 2022-07-11

**Rating:** 5
**Confidence:** 3
**Soundness:** 3 good
**Presentation:** 3 good
**Contribution:** 2 fair

**Summary:**

This paper studies a sequential bilateral trade problem, where one seller and one buyer arrive at every round. The seller and the buyer both have a private valuation of the item that are adversarial. The learning algorithm posts one or two prices for the seller/buyer and aims to minimize the alpha-regret of the trade over T rounds (alpha-regret is due to the hardness result shown by prior work).

The paper considered different types of feedbacks including the full feedback (seller and buyer's values are revealed); the one-bit feedback (only the trade happened or not is revealed), and the two-bit feedback (whether the seller or the buyer accepts the price). The paper also studied these feedbacks with different strategy of the learner --posting one price or two prices.

The main results include a hardness result which shows that no algorithm can achieve sublinear alpha-regret when alpha<2. Then, with alpha=2, the authors provide sublinear regret algorithms with two-price strategy, one-price strategy under different types of feedbacks. When the feedback is incomplete (one-bit or two-bit), two-price algorithms are needed in order to achieve a sublinear regret rate.

**Questions:**

(please see the above weakness section too)
- about the gaps in the current results: under the two-bit feedback system can the two-price algorithm achieve T^{2/3} regret upper bound, for example by an explore-then-commit type strategy with the discretization?
- Is it necessary for the Block-Decomposition algorithm to know T, can one apply the doubling trick and make it independent of T?
- the paper proposed the estimation procedure to estimate the price unbiasedly -- would it be useful to apply an upper/lower confidence bound for the prices over the rounds (or reasons that was not needed/used) ?

**Limitations:**

Yes (work is mainly theoretical)

**Strengths And Weaknesses:**

Strength:
- the paper is well-written and I appreciate that the authors provided a detailed review on the prior works, and how the setup and techniques in this work are different from those.
- I found the technical part of the paper to be interesting, including the estimation procedure using two prices and the lower bound example 's construction.

Weaknesses:
- the upper bound results were a bit less exciting-- after doing the discretization, one can then use the prediction with experts framework on a grid of prices.
- the results in table 1 contains a few gaps, in particular the study under the one-bit feedback seems to be quite incomplete and many questions remain open. e.g. what is the best possible regret a one-price strategy can achieve with one-bit feedback? what about the two-price strategy?

---

> ### Author Response · Authors · 2022-08-02
> **Our Response to Reviewer VWf6**
>
> $\bf{Weaknesses}$
>
> - the upper bound results were a bit less exciting-- after doing the discretization, one can then use the prediction with experts framework on a grid of prices.
>
> [Answer]: This is indeed true for the full feedback, but in the partial feedback models we need two extra ingredients to apply a reduction to experts: a block decomposition of the time horizon and a procedure to estimate the gain from trade using only one (or two bits feedback). While the block decomposition is a common trick in learning with partial feedback, the design of the estimator is quite delicate and it was surprising to us.
>
> - the results in table 1 contains a few gaps, in particular the study under the one-bit feedback seems to be quite incomplete and many questions remain open. e.g. what is the best possible regret a one-price strategy can achieve with one-bit feedback? what about the two-price strategy?
>
> [Answer]: The gaps in Table 1 are misleading because upper bounds in weaker models apply in stronger models and lower bounds in stronger models apply in weaker models. In particular,  a lower bound for two bit feedback is also a lower bound for the one bit feedback, hence our results implies a linear lower bound to the 2-regret for one-price strategy with one-bit feedback. In addition, a lower bound for two price strategies is also a lower bound for the single price ones. The remaining open gap in our results is between the $T^{2/3}$ lower bound and the $T^{3/4}$ upper bound that hold for two prices and partial feedback (either 1 or 2 bit feedback).
>
> $\bf{Questions:}$
>
> - about the gaps in the current results: under the two-bit feedback system can the two-price algorithm achieve $T^{2/3}$ regret upper bound, for example by an explore-then-commit type strategy with the discretization?
>
> [Answer]: Explore-then-commit strategies are mainly relevant in stochastic (e.g., i.i.d.) scenarios. Our work is in the adversarial model and there is no relation between past and future agents' valuations.
>
> - Is it necessary for the Block-Decomposition algorithm to know T, can one apply the doubling trick and make it independent of T?
>
> [Answer]: It is straightforward to extend our learning algorithms to achieve the same asymptotic bounds even when the time horizon is not known up-front by using the “standard” doubling trick.
>
> - the paper proposed the estimation procedure to estimate the price unbiasedly -- would it be useful to apply an upper/lower confidence bound for the prices over the rounds (or reasons that was not needed/used) ?
>
> [Answer]: Our estimation procedure exploits the randomness of the pricing strategy (that we control) to extract information from a deterministic but unknown adversarial sequence. We are in the adversarial case and there is no underlying distribution that we seek to learn over time, thus the confidence measure is inapplicable in our setting.
>
> $\bf{Limitations:}$
> - Yes (work is mainly theoretical)
>
> [Answer]: Please note that “Theory (e.g., control theory, learning theory, algorithmic game theory)” is one of the specific topics appearing in the call for papers of NeurIPS. In addition,  our algorithms are easy to implement and are not only for theoretical purposes.

---

> > ### Comment · Reviewer_VWf6 · 2022-08-08
> > **reply to author response**
> >
> > Dear authors,
> >
> > Thanks for the detailed reply and that answered my questions.

---

### Official Review · Reviewer_4zj6 · 2022-07-12

**Rating:** 8
**Confidence:** 4
**Soundness:** 4 excellent
**Presentation:** 4 excellent
**Contribution:** 4 excellent

**Summary:**

The paper studies sequential bilateral trade where sellers and buyers valuations are adversarially chosen. The paper considers two types of posted price mechanisms: (1) mechanisms that post a single price and (2) mechanisms that post different prices for buyer and seller. In addition, three types of feedback are considered: full observation of the seller’s and the buyer’s valuations, observing whether the trade happens, and observing whether the seller and the buyer are willing to trade respectively. The paper proves both lower bounds and upper bounds of the regret when the performance is measured by gain from trade. It first gives a strong impossibility result that no mechanism can achieve sublinear a-regret for any a<2, so 2-regret is the best we can hope for. Second, with full feedback sublinear 2-regret is proved to be achievable. Third, for partial feedback, they show a contrast between single-price mechanisms and two-price mechanisms: with a single price, one cannot get sublinear a-regret for any constant a; but posting two prices even with one-bit feedback achieves sublinear 2-regret. Finally, they prove that there is a separation in the 2-regret bounds between full and partial feedback.

**Questions:**

I have no questions.

**Limitations:**

Yes.

**Strengths And Weaknesses:**

The paper studies an important problem and proves strong results. The fact that setting two prices can drastically improve the performance in the sequential setting is fascinating to me. I find no reason to reject the paper.

---

### Author Response · Authors · 2022-08-02
**To all the Reviewers**

We sincerely thank all the reviewers for their time and constructive feedback.
Below, we address the issues raised by answering separately to each review, where needed.
For clarity of exposition, we report in bullets the parts of the reviews we address.

---

### Meta-Review · Area_Chair_cGBn · 2022-09-01

**Recommendation:** Accept
**Confidence:** Certain

**Metareview:**

This is an interesting paper on bandit with non-trivial structure and algorithms.

All the reviews are positive, as is my own opinion.

I quite happily suggest acceptance for this paper.

**Award:**

No

---

### Decision · Program_Chairs · 2022-09-14

Accept